# A non-canonical repressor function of JUN restrains YAP activity and liver cancer growth

Yuliya Kurlishchuk[1,2], Anita Cindric Vranesic[1,2], Marco Jessen [ID][1,2], Alexandra Kipping[1], Christin Ritter[1], KyungMok Kim[1], Paul Cramer[1] & Björn von Eyss [ID][1✉]

## Abstract

Yes-associated protein (YAP) and its homolog, transcriptional coactivator with PDZ-binding motif (TAZ), are the main transcriptional downstream effectors of the Hippo pathway. Decreased Hippo pathway activity leads to nuclear translocation of YAP/TAZ where they interact with TEAD transcription factors to induce target gene expression. Unrestrained YAP/TAZ activity can lead to excessive growth and tumor formation in a short time, underscoring the evolutionary need for tight control of these two transcriptional coactivators. Here, we report that the AP-1 component JUN acts as specific repressor of YAP/TAZ at joint target sites to decrease YAP/TAZ activity. This function of JUN is independent of its heterodimeric AP-1 partner FOS and the canonical AP-1 function. Since expression of JUN is itself induced by YAP/TAZ, our work identifies a JUN-dependent negative feedback loop that buffers YAP/TAZ activity at joint genomic sites. This negative feedback loop gets disrupted in liver cancer to unlock the full oncogenic potential of YAP/TAZ. Our results thus demonstrate an additional layer of control for the interplay of YAP/TAZ and AP-1.

Keywords AP-1; Hippo; Liver Cancer; JUN; YAP
Subject Categories Cancer; Chromatin, Transcription & Genomics

## Introduction

The transcriptional coactivators YAP/TAZ are the critical downstream regulators of the Hippo pathway that regulate gene expression in response to changes in pathway activity, mainly by binding to TEAD transcription factors (Dong et al, 2007; Zhao et al, 2007). Uncontrolled transcriptional output of YAP/TAZ can lead to rapid induction of aggressive tumor growth, e.g., in the liver (Moya et al, 2019; Wu et al, 2022; Yimlamai et al, 2014; Zhou et al, 2009). For this reason, it is imperative for an organism to ensure tight and well-orchestrated control over YAP/TAZ to protect the body from the fatal consequences of their derailed activity (Driskill and Pan, 2021).

AP-1 is a dimeric basic leucine zipper transcription factor complex with JUN and FOS proteins being the most abundant members of this family (Eferl and Wagner, 2003). Unlike FOS, JUN can also form homodimers, but in the cell, JUN preferentially forms heterodimers with members of the FOS family, which act as potent transcriptional activators (Vogt, 2002). Previous studies identified substantial co-occupancy of YAP/TAZ and AP-1 at genomic sites, and they demonstrated that JUN/FOS heterodimers cooperate with YAP/TAZ and TEAD transcription factors to drive YAP/TAZ target gene expression (Koo et al, 2020; Shao et al, 2014; Zanconato et al, 2015).

A complete understanding of all YAP/TAZ control mechanisms may thus provide a basis for novel cancer therapies. Furthermore, targeting YAP/TAZ given their pivotal roles in regeneration (Elster and von Eyss, 2020; Leach et al, 2017), holds great promise for enhancing this process.

In this article, we now elucidate a negative feedback mechanism in which high YAP activity is restrained by the recruitment of JUN/NCOR1 repressor complexes and show that this non-canonical JUN function is part of a tumor suppressor mechanism in the liver.

## Results

### JUN antagonizes YAP5SA-mediated growth arrest

We observed that expression of the Hippo-insensitive hyperactive YAP5SA mutant (Zhao et al, 2007) at endogenous levels in MCF10A cells with a doxycycline-inducible YAP5SA (iYAP5SA) resulted in greatly increased proliferation, whereas strong constitutive lentiviral overexpression of the YAP5SA allele in MCF10A cells resulted in markedly reduced growth of these cells (Fig. 1A,B). This reduced growth phenotype was TEAD-dependent since adding the S94A TEAD-binding mutation to the YAP5SA allele completely abolished the effect on cell growth (Fig. 1A,B). The detrimental effect of supraphysiological YAP5SA expression levels was specific to growth in 2D conditions, since mammosphere formation was enhanced (Fig. 1C) as described previously (von Eyss et al, 2015). Given that stable overexpression of YAP5SA over a period of several days led to a significant reduction in growth (Fig. 1D), we leveraged this overexpression phenotype as a selective pressure in a screening approach. In particular, we aimed to screen for genes that can counteract this phenotype when overexpressed, potentially allowing us to identify

---

[1]Transcriptional Control of Tissue Homeostasis Lab, Leibniz Institute on Aging, Fritz Lipmann Institute e.V., Beutenbergstr. 11, 07745 Jena, Germany. [2]These authors contributed equally: Yuliya Kurlishchuk, Anita Cindric Vranesic, Marco Jessen. ✉E-mail: bjoern.voneyss@leibniz-fli.de

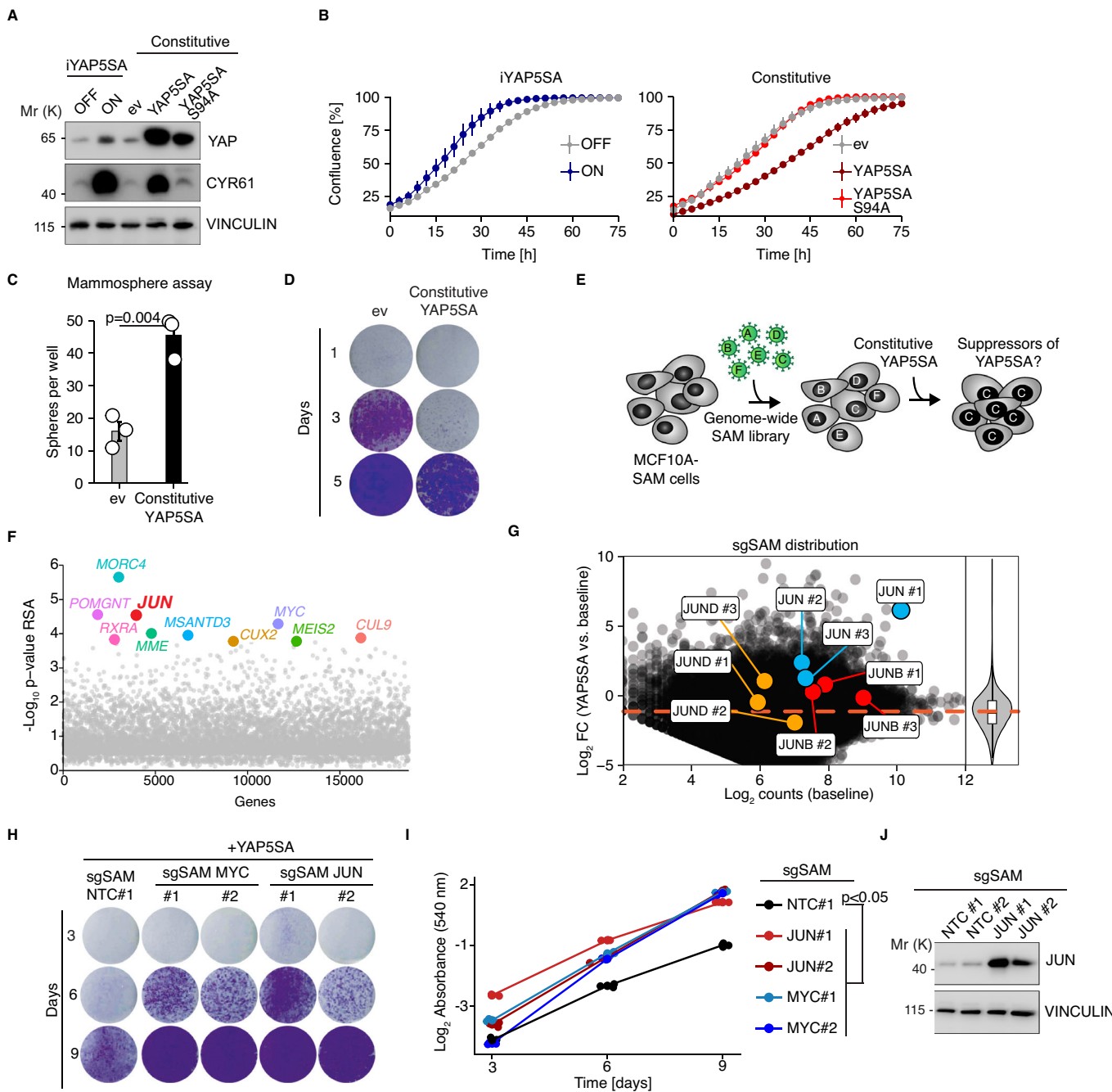

**Figure 1. JUN antagonizes YAP5SA-mediated growth arrest.**

(A) Immunoblot from MCF10A cells infected with the indicated lentiviral vectors. iYAP5SA MCF10A cells were treated with either EtOH as a solvent control (iYAP5SA OFF) or doxycycline (iYAP5SA ON), $n = 3$ (biological replicates). (B) Incucyte growth curves for the indicated MCF10A cell lines, $n = 3$ (biological replicates). (C) Quantification of mammosphere assays for the indicated MCF10A cells, $n = 3$ (biological replicates). (D) Crystal violet stain for the indicated MCF10A cell lines, $n = 2$ (biological replicates). (E) Schematic illustrating the SAM screen to identify suppressors of YAP5SA-mediated growth arrest. (F) Summary of SAM screen for enrichment of sgSAMs (14 days after YAP5SA overexpression/baseline) targeting all human genes. The redundant siRNA algorithm (RSA) was used to integrate all sgSAMs targeting one gene and to infer statistical significance of enrichment, $n = 1$. (G) MA plot illustrating the distribution of sgSAMs targeting JUN family members. The violin plot (right) illustrates the Log2 fold changes (YAP5SA vs. baseline) for all sgRNAs that are plotted in the MA plot. (H) Crystal violet of MCF10A SAM cells expressing individual sgSAMs which were superinfected with YAP5SA and stained after 3, 6, and 9 days of YAP5SA infection. $n = 3$ (technical replicates). (I) Quantification of crystal violet from (G). Two-way ANOVA. (J) Immunoblot from MCF10A SAM cells infected with the indicated JUN and control sgSAMs. NTC non-targeting control, $n = 3$ (biological replicates). The error bars in this figure indicate the standard error of the mean. Source data are available online for this figure.

candidates that act as suppressors of ectopic (supraphysiological) YAP activity. We used the genome-wide *Synergistic Activation Mediator* (SAM) library (Konermann et al, 2015) to identify genes that suppress the YAP5SA-dependent growth defect when overexpressed. MCF10A-SAM cells infected with a genome-wide sgSAM library were subsequently superinfected with constitutive YAP5SA. Cells were kept in culture for two weeks to allow outgrowth of cells expressing potential suppressors of supraphysiological YAP5SA levels (Fig. 1E). Our analyses identified numerous genes that were significantly enriched in the screen (Fig. 1F, Dataset EV1), including MYC among the top hits, which was previously described as a Hippo pathway-independent suppressor of YAP (Croci et al, 2017; von Eyss et al, 2015). *JUN* (c-JUN) particularly piqued our interest because several studies proposed a cooperative behavior between YAP and AP-1 in cancer cells (Koo et al, 2020; Stein et al, 2015; Zanconato et al, 2015). Our data, however, would argue that JUN suppresses YAP function. For this reason, we followed up more closely on this interesting but seemingly counterintuitive finding. In our SAM screen, *JUN* was the only member of the JUN family that showed an enrichment of sgSAMs, while *JUND* and *JUNB* sgSAMs were not enriched (Fig. 1G). We validated the results from our SAM screen using MCF10A-SAM cells infected with individual sgSAMs (Fig. 1H–J). Like in the SAM screen, sgSAM JUN#1 was more potent than sgSAM JUN#2 in terms of its ability to rescue cell growth (Fig. 1H,I), which correlated with JUN expression on protein level (Fig. 1J). JUN expression itself (in the absence of strong constitutive YAP5SA overexpression) did not have a major impact on proliferation (Appendix Fig. S1A–C).

## JUN interferes with induction of a large fraction of YAP target genes

We next analyzed the transcriptome of JUN-overexpressing (sgSAM JUN#1) MCF10A-SAM cells. Gene set enrichment analysis (GSEA) showed that YAP target genes were the most potently downregulated gene set after JUN overexpression (Fig. 2A,B). These results could be corroborated with cDNA-mediated expression of JUN since a mild JUN overexpression led to induction of the AP-1 target gene *IL1B* and to potent downregulation of *ANKRD1* and *THBS1* mRNA expression (Fig. 2C,D), both part of the Cordenonsi YAP gene set. In addition, JUN was able to suppress YAP-dependent induction of the *ANKRD1* promoter in a reporter assay (Fig. EV1A,B). To investigate a potential involvement of the Hippo pathway in this, we used the iYAP5SA MCF10A cells expressing the YAP5SA allele at endogenous levels in a doxycycline-dependent manner. First, we performed independent RNA-Sequencing (RNA-Seq) experiments applying a short doxycycline treatment to enrich for direct YAP target genes and limit secondary effects (Fig. EV2A,B, Dataset EV2). Consistently, we saw upregulation of previously published conserved YAP gene signatures (Fig. EV2C,D). We then inferred the Top-YAP-induced (TYI) genes (log2FC < 1, padj < 1e−4, *n* = 434 genes, Dataset EV2) from this experiment for our consecutive analyses. Next, iYAP5SA cells were infected with a JUN expression vector or a vector control (Fig. 2E). Consistent with previous reports (Maglic et al, 2018), YAP was able to induce JUN expression at the protein level, similar to JUN overexpression by cDNA (Fig. 2E). Subsequent RNA-Seq and unsupervised clustering analyses (Fig. 2F,G) revealed three main clusters (Dataset EV3) within TYI genes: cluster 3 genes mainly consisted of known AP-1-induced target genes, cluster 1 and cluster 2 contained well-established direct YAP/TAZ target genes (Fig. 2G,H).

Whereas JUN overexpression had only a very weak or no effect on cluster 2 genes, YAP-dependent induction of cluster 1 genes was completely blunted by JUN overexpression (Fig. 2F–I). The inhibitory effect on YAP target gene induction was not an exclusive feature of JUN because JUNB was also able to repress YAP target genes whereas JUND was strongly attenuated in this respect despite its high expression levels (Fig. 1J,K). In addition, JUN was able to specifically blunt the induction of cluster 1 genes (Fig. 2L,M) triggered by endogenous YAP/TAZ activation upon treatment with the LATS kinase inhibitor TRULI (Kastan et al, 2021).

Next, we performed SLAM-Seq experiments (Fig. 2N) in conjunction with acute JUN protein depletion, as described previously for MYC (Muhar et al, 2018). MCF10A JUN knockout (KO) cells were reconstituted with a JUN allele fused to a V5-tagged auxin-inducible degron (JUN-AID-V5) which was rapidly degraded after 1 h of indole-3-acetic acid (IAA) treatment (Fig. 2O; Appendix Fig. S2A,B). Based on the T-to-C conversions in SLAM-Seq that occur in de novo synthesized mRNAs at the 4-thiouridine (4sU) sites, one can distinguish between changes at the steady-state level, and changes in de novo synthesized mRNA during the 4sU pulse/JUN degradation phase (Fig. 2P). Whereas there were no significant changes in the fraction of total mRNAs of YAP/TAZ target genes (Dataset EV4), these target genes were significantly induced in the fraction of de novo mRNAs (Fig. 2P) arguing for a direct effect of JUN on YAP target genes, and against secondary mechanisms, e.g., via the JUN-dependent induction of a repressor acting on YAP target genes.

## JUN is recruited to weak enhancer sites after YAP induction

Based on the results of the SLAM-Seq, we hypothesized that JUN directly interferes with YAP/TEAD on chromatin. To identify the sites bound by YAP/TEAD and JUN/FOS, we performed CUT&RUN experiments from MCF10A cells constitutively overexpressing YAP5SA (Figs. 3A and EV3A,B) as well as from cells expressing iYAP5SA infected with JUN expression vectors (Fig. 3B–I). Consistent with previous results in cancer cell lines, JUN and FOS bound to many YAP/TEAD target sites (joint Y/T AP-1), but they also bound to AP-1 exclusive sites (AP-1 only), where no YAP/TEAD could be detected (Fig. 3A). AP-1-related motifs were strongly enriched in both YAP and TEAD1 peaks (Fig. EV3B). We next investigated how JUN binding is affected by acute YAP induction. It should be mentioned here that this analysis is complicated by the fact that JUN itself is induced by YAP, so that one can also observe effects that simply occur due to increased JUN protein levels and are not necessarily mediated by recruitment of YAP to genomic sites.

Whereas JUN overexpression in uninduced conditions (JUNoe; iYAP5SA OFF) led to 1203 additional JUN sites (Fig. 3B), YAP induction in empty vector-infected cells (ev; iYAP5SA ON) led to 6999 unique sites, indicating that these YAP-dependent sites cannot be simply explained by increased JUN protein levels.

YAP induction itself led to a potent recruitment of YAP and JUN at joint YAP/TEAD sites (*n* = 16,307 sites, Fig. 3C,D). While recruitment of YAP was largely unaffected by either JUN overexpression or the presence of an AP-1 binding motif at these sites, recruitment of JUN was more pronounced in peaks with an AP-1 motif (Fig. 3D). This indicates that DNA-binding of JUN contributes to YAP-dependent recruitment. We next performed a differential peak analysis to identify peaks that quantitatively change after YAP induction (iYAP5SA ON vs. iYAP5SA OFF). Here, we analyzed the YAP and JUN signals at a

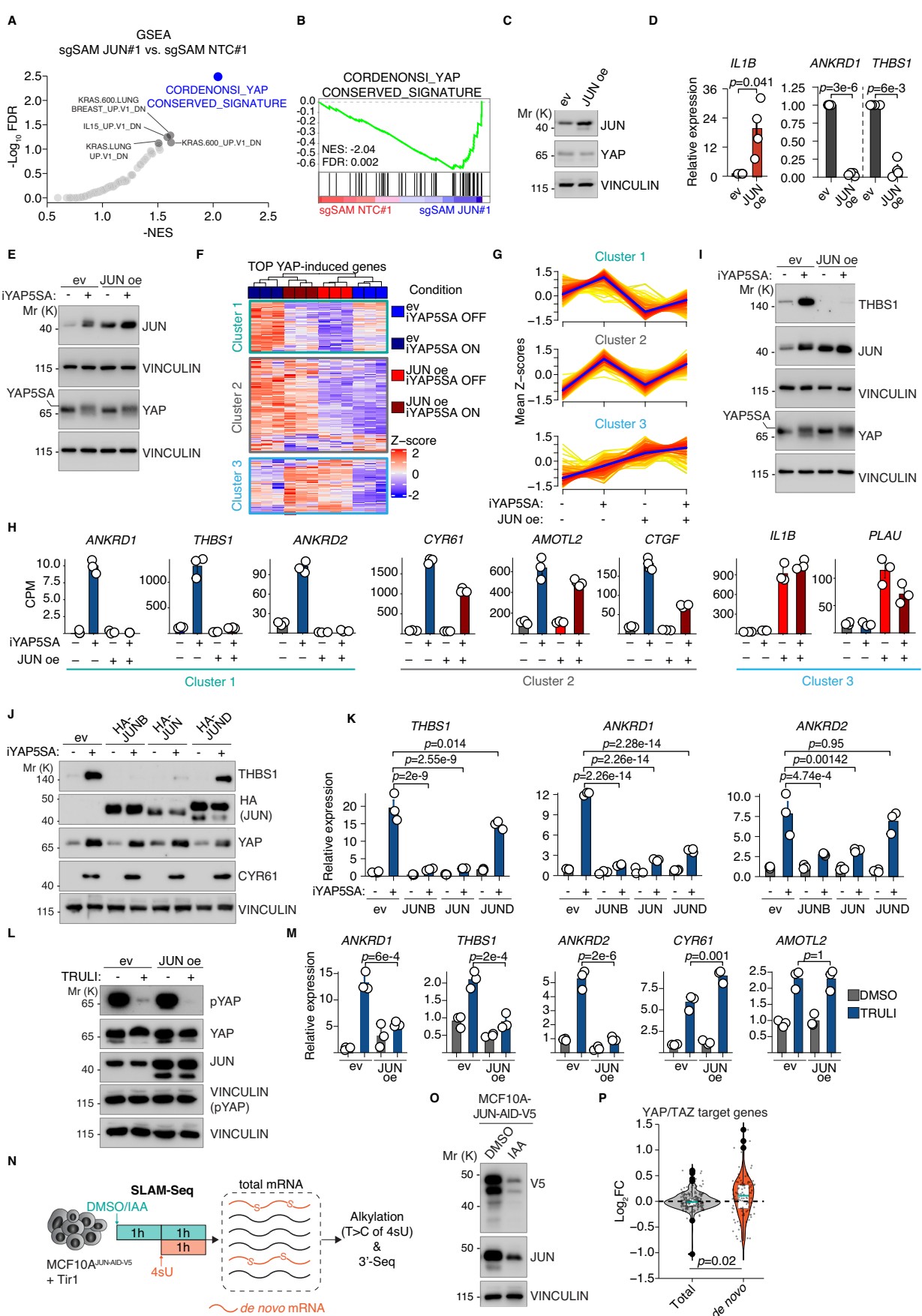

**Figure 2. JUN proteins interfere with YAP target gene expression.**

(A) GSEA summary of downregulated gene sets from RNA-Seq data comparing sgSAM JUN#1 vs. sgSAM NTC#1 cells. NES normalized enrichment score, FDR false discovery rate, $n = 2$ (biological replicates). (B) GSEA enrichment plot for the Cordenonsi gene signature as the most strongly downregulated gene set in sgSAM JUN#1. NES normalized enrichment score, FDR false discovery rate. (C) Immunoblot of MCF10A cells infected with JUN cDNA vectors or an empty vector control (ev). (D) qRT-PCR analysis of MCF10A-SAM cells after JUN cDNA overexpression. Summary of four biological replicates. Welch T-test. (E) Immunoblot analysis of iYAP5SA cells infected with JUN overexpression vectors or an empty vector control (ev). YAP5SA was induced for 16 h and subsequently analyzed, $n = 4$ (biological replicates). (F) Heatmap of TOP YAP-induced genes (Z-score normalized, $n = 434$ genes) in the indicated experimental conditions in iYAP5SA cells, analyzed by RNA-Seq and subsequent unsupervised clustering with three clusters. $n = 3$ biological replicates per experimental group. (G) Summary of mean Z-scores (per experimental group) for all TOP YAP-induced genes to illustrate the expression changes in the four experimental groups. (H) Barplots illustrating the expression of representative genes for each cluster in the RNA-Seq experiment ($n = 3$, biological replicates). CPM mapped counts per million. (I) Immunoblot analysis of iYAP5SA cells that were infected with JUN overexpression vectors or an empty vector control (ev). YAP5SA was induced for 16 h and subsequently analyzed. (J) Immunoblot analysis of iYAP5SA cells that were infected with JUN overexpression vectors or an empty vector control (ev). YAP5SA was induced for 24 h and subsequently analyzed, $n = 2$ (biological replicates). (K) qRT-PCR analysis of the same conditions as in (J) ($n = 3$, biological replicates). One-way ANOVA. (L) Immunoblot analysis of MCF10A cells that were treated with 10 μM TRULI or DMSO control for 24 h. (M) qRT-PCR analysis of the same cells from (L) ($n = 3$, biological replicates). One-way ANOVA. (N) Schematic of the SLAM-Seq approach that combines acute JUN depletion via the auxin system with metabolic labeling of de novo mRNA transcripts using 4-thiouridine (4sU) in MCF10A$^{JUN-AID-V5}$ cells. IAA indole-3-acetic acid. (O) Immunoblot from MCF10A$^{JUN-AID-V5}$ cells 2 h after indole-3-acetic acid (IAA) addition to induce JUN degradation. (P) Violin plots of YAP/TAZ target genes that illustrate the gene expression changes upon acute JUN depletion for the total mRNA pool (left) versus the de novo transcribed pool (right). Two-sided Wilcox test ($n = 3$, biological replicates). The error bars in this figure indicate the standard error of the mean. The boxplots encompass data points between the first and third quartiles. The median is indicated by a horizontal line. Whiskers extend to 1.5x interquartile range; any data points outside that range are shown as individual dots. Source data are provided as a Source data file. Source data are available online for this figure.

merged peak set comprising all 14,651 YAP peaks detected in any of the four conditions (Fig. 3B). We were able to identify 1752 differentially bound regions (DBRs) for YAP and 4381 DBRs for JUN (Fig. 3E). Next, we stratified the YAP peaks based on falling into one of the following DBR category: "YAP only", for "JUN only" or for "YAP and JUN" (Fig. 3F). The distance of "YAP only" DBRs to the nearest transcriptional start site (TSS) tended to be significantly smaller to TSSs then the other DBR categories (Fig. 3G). Consistently, the distance of YAP peaks to the TSSs of cluster 2 genes—which are transcriptionally regulated by YAP but not JUN—was significantly smaller compared to cluster 1 genes (Fig. 3H).

Since this analysis pointed towards a specific function of JUN at distant YAP-regulated enhancers, we performed additional CUT&RUN experiments to map the enhancer landscape in MCF10A cells. To this end, we performed CUT&RUN experiments for enhancer-specific histone modifications, namely H3K4me1 and H3K27ac and integrated the data with published iYAP5SA MCF10A ATAC-Seq data (Fetiva et al, 2023). After data integration, we were able to identify 41,736 putative enhancer sites in MCF10A cells (Fig. EV3C). When overlaying these data with our YAP and JUN CUT&RUN data (Fig. 3I–M), we could not observe major differences between the binding behavior of YAP and JUN when comparing "iYAP5SA only" JUN peaks that are recruited in a strictly YAP-dependent manner (highlighted in green in Fig. 3) vs. "Joint" JUN peaks that are detected in all four conditions (highlighted in yellow in Fig. 3). The only exception was that JUN was already present at joint JUN peaks under uninduced "iYAP5SA OFF" conditions, which is consistent with the prior selection of this peak set (Fig. 3I). Notably, those enhancers overlapping with "iYAP5SA only" JUN peaks showed a significantly lower signal for H3K4me1, H3K27ac (Fig. 3J) and chromatin accessibility (Fig. 3K) compared to enhancers located in "joint" JUN peaks. This suggests that these YAP-dependent enhancers are rather weak enhancers in an uninduced state. However, these weak enhancer sites show potent recruitment of YAP and JUN (Fig. 3I) as well as a strong increase in chromatin accessibility (Fig. 3K,M) indicative of an increased enhancer function. Thus, these data imply that JUN is able to specifically restrict YAP activity at weak

enhancers, potentially in order to restrict enhancer invasion induced by oncogenic levels of YAP.

## Canonical AP-1 function and YAP inhibition are distinct JUN properties

T-5224 is an inhibitor that interferes with AP-1 function by binding AP-1's basic region required for DNA binding (Aikawa et al, 2008; Tsuchida et al, 2004). Furthermore, T-5224 can inhibit with growth of YAP-dependent liver tumors (Koo et al, 2020). We tested whether T-5224 would specifically impact JUN-dependent YAP targets. Despite JUN induction on protein level (Fig. 4A), T-5224 reduced the expression of canonical AP-1 targets such as *IL1B* or *PLAU* (Fig. 4A,B). In addition, cluster 1 genes were potently downregulated, whereas the expression of cluster 2 genes barely changed (Fig. 4B). One reason for T-5224's ability to downregulate cluster 1 genes could be the fact that FOS can enhance expression of YAP target genes (Koo et al, 2020), but that alternative JUN-containing AP-1 complexes exist, which in turn mediate the repression of YAP target genes. However, enforced expression of FOS was not sufficient to induce expression of cluster 1 genes in conjunction with YAP5SA induction (Fig. EV4A,B) suggesting that FOS expression alone is not sufficient to promote the formation of activating complexes. Yet, the ability of JUN to limit the transcriptional activity of YAP appears to be distinct from its canonical function in AP-1, i.e., induction of JUN/FOS target genes such as *IL1B*.

To test this, we set out to find JUN mutants that would allow us to separate these two different JUN functions. Previous work identified leucine zipper mutants of JUN that showed differential activities towards activating AP-1-dependent reporter genes, potentially due to different abilities to form specific AP-1 homo- and heterodimers (Smeal et al, 1989). Here, we tested two mutants: JUN M14 and JUN I10. The M14 mutant harbours three point mutations in the leucine zipper region and retains the ability to bind DNA, whereas the insertion of four amino acids into the leucine zipper region of the JUN I10 mutant abolishes DNA binding (Fig. 4C). JUN WT overexpression in MCF10A cells induced IL1B expression and downregulated THBS1 on protein level, while JUN I10 did neither affect induction of IL1B nor led to downregulation of THBS1 (Fig. 4D). JUN M14, however, was

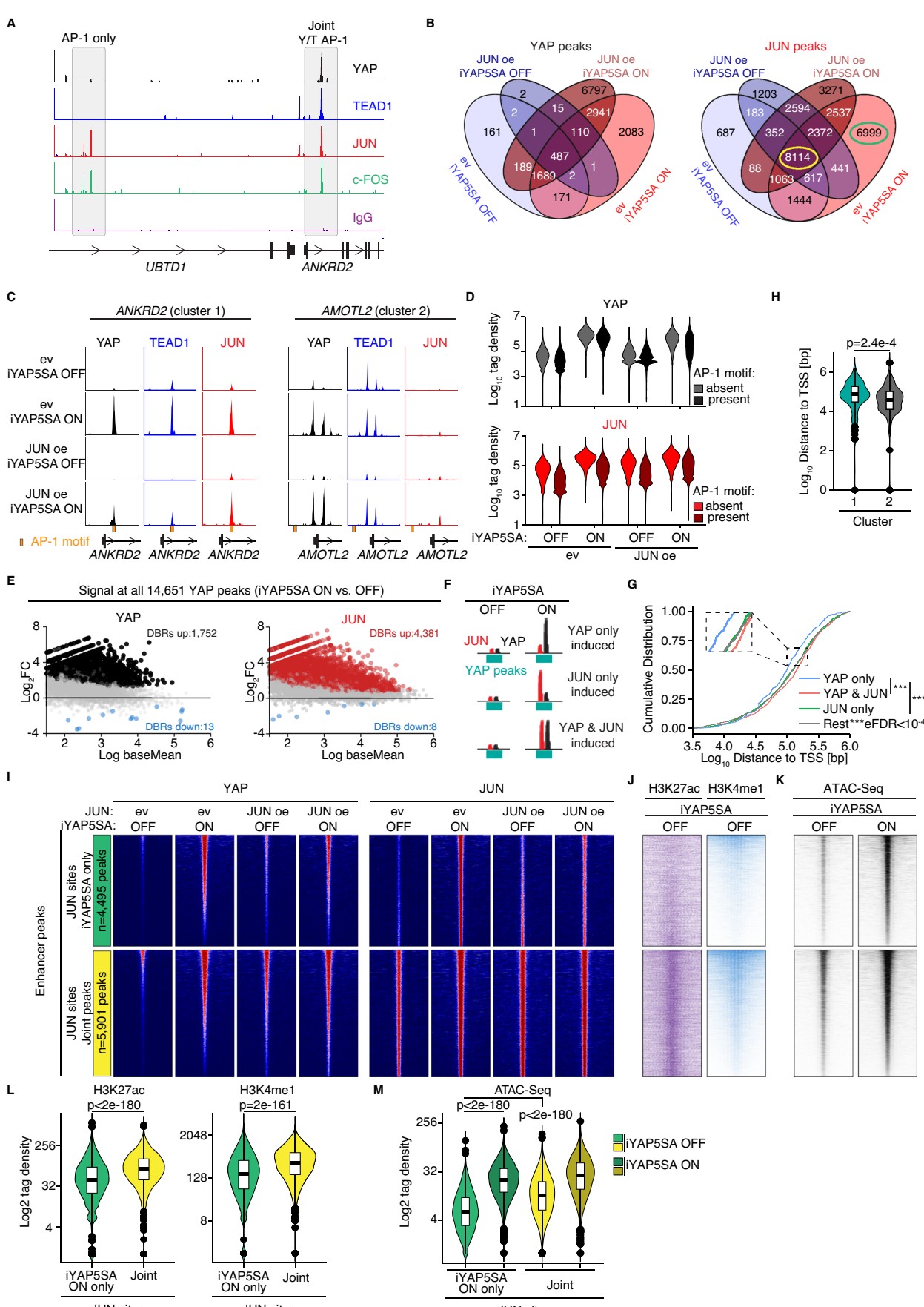

**Figure 3.   JUN is recruited by YAP to genomic sites.**

(A) Representative CUT&RUN sequencing tracks of MCF10A stably overexpressing YAP5SA that illustrate binding of AP-1 only (JUN and FOS, left) and joint YAP/TEAD AP-1 (Y/T AP-1, right) sites. (B) Venn diagram for YAP and JUN CUT&RUN peaks in MCF10A-iYAP5SA cells infected with a JUN expression construct or empty vector control (ev). One day prior to CUT&RUN, cells were treated with either EtOH as a solvent control (iYAP5SA OFF) or doxycycline (iYAP5SA ON). (C) Representative CUT&RUN sequencing tracks that illustrate the binding to *ANKRD2* and *AMOTL2*. AP-1 DNA binding motifs are highlighted with an orange box. (D) Count distribution within 200 bp windows of 16,307 joint YAP/TEAD peaks (see Fig. EV3A) stratified based on the presence of an AP-1 motif in the peak ($n = 3$, biological replicates). (E) MA plots for the YAP and JUN signal, respectively, at all YAP peaks comparing the YAP5SA ON vs. YAP5SA OFF condition. Significantly induced regions are highlighted in color. DBR differentially bound region. (F) Schematic to stratify the peaks based on their differential binding behavior at all 14,561 YAP peaks upon YAP5SA induction. (G) Cumulative distribution frequency for the distance of peaks that were stratified as described in (F). Each cumulation distribution frequency was tested against 1e4 random gene sets of similar size using a Kolgomorov–Smirnov test. eFDR empirical false discovery rate; ***≤1e−4. (H) Distance of TSSs from cluster 1 and cluster 2 genes to the nearest YAP peak. Two-sided Wilcox test ($n = 3$, biological replicates). (I) CUT&RUN heatmaps for binding of YAP, TEAD1, and JUN ($n = 3$, biological replicates). Enhancer sites were overlapped with "YAP5SA only" JUN peaks that occur in a YAP-dependent manner (exclusively in the ev; YAP5SA ON condition, highlighted in green, $n = 4495$ peaks) or a control peak set of similar size ("Joint peaks", highlighted in yellow, $n = 5901$ peaks). See also (B) for the JUN peaks that were used here. All heatmaps were sorted based on the YAP signal in the ev; YAP5SA ON situation. (J) CUT&RUN heatmaps for the enhancer marks H3K4me1 and H3K27ac ($n = 3$, biological replicates) at the same sites as in (I). (K) ATAC-Seq (taken from GSE193704) heatmaps ($n = 2$ per condition) for the same sites as in (I). (L, M) Violplots for tag densities for the heatmaps in (J, K). Two-sided Wilcox-test. All boxplots encompass data points between the first and third quartiles. The median is indicated by a horizontal line. Whiskers extend to 1.5x interquartile range; any data points outside that range are shown as individual dots.

able to potently decrease expression of cluster 1 genes, whereas expression of *IL1B, CXCL8,* and *CYR61* remained largely unchanged (Fig. 4D,E). Since JUN M14 showed a clear difference in terms of canonical AP-1 functions versus repression of YAP target genes, we analyzed this mutant in more detail, as it would allow us to separate the two JUN functions. CUT&RUN experiments for JUN in JUN KO cells that were reconstituted with either JUN WT, JUN M14 or a vector control showed that JUN M14 had a similar binding behavior as JUN WT (Fig. 4F,G). Furthermore, we were not able to identify any consistent differences in their binding behavior towards joint YAP/ JUN targets or JUN only targets, even though JUN M14 tended to show a stronger signal (Fig. 4H,I) which could be simply due to higher expression levels (Fig. 4D). Thus, JUN M14 is unable to activate canonical AP-1 target genes but retains the ability to bind to genomic targets and interfere with YAP function.

## JUN M14 shares a similar interactome with JUN WT but is deficient in heterodimerization with FOS

To identify critical mediators of JUN-dependent repression, we performed BioID experiments in MCF10A JUN KO cells reconstituted with different Flag-BirA* JUN fusion proteins or nuclear localization signal (NLS)-Flag-BirA* control (Fig. 5A). In the mass spectrometry analysis (see Dataset EV5 for a list of all interactors) all JUN fusion proteins showed a strikingly similar interactome in the principal component analysis and the overlap of all interactors (Fig. 5B,C). Here, a strong enrichment for components of SWI/SNF complexes and proteins involved in hepatocellular carcinogenesis could be observed (Fig. 5D). Cluster analysis of interactors showed that ATP-dependent chromatin remodeling components (e.g., ARID1A and SMARCA2/4) and regulators of RNA polymerase (RNAP) II (e.g., NCOR1/2) were high confidence interactors of JUN WT and JUN M14 (Fig. 5E,F). We also identified YAP as a JUN WT/M14 interactor, confirming our hypothesis that JUN is recruited to shared YAP/JUN sites via a protein interaction with YAP. When comparing JUN WT with JUN M14, all FOS-like proteins (e.g., FOS, FOSB, FOSL2) showed at least a 4-fold reduced labeling efficiency in JUN M14 (Fig. 5G). This was confirmed by co-immunoprecipitation, as FOS and FRA2 were significantly reduced in Flag precipitates from BirA*-Flag JUN M14 compared with BirA*-Flag JUN WT (Fig. 5H). On the other hand, JUN M14 was still able to homodimerize with JUN WT, which even tended to be more

pronounced in JUN M14 compared with JUN WT (Fig. 5I). The formation of JUN::JUN homodimers could partially be inhibited by enforced FOS expression which was paralleled by JUN::FOS heterodimer formation, demonstrating that homo- vs. heterodimers are exclusive JUN complexes (Fig. EV4C). This suggests that the impaired function of JUN M14 to induce canonical AP-1 targets is due to its reduced interaction with FOS proteins, whereas the effect on YAP targets is largely FOS-independent.

## NCOR1/2 are required for JUN's ability to repress YAP target genes

To identify proteins that could mediate the repressive function of JUN, we focused on the SWI/SNF component ARID1A/B and the corepressor protein NCOR1/2 since both can behave as repressors of transcription and were potently enriched for all JUN proteins (Fig. 5E,F). To this end, we depleted ARID1A/B and NCOR1/2 in iYAP5SA cells overexpressing JUN WT by siRNA. Whereas ARID1A/B-depleted cells behaved like control-depleted cells, NCOR1/2 depletion was able to completely restore THBS1 in JUN-overexpressing cells under YAP5SA-induced conditions (Fig. 6A; Appendix Fig. S3A). In addition, the knockout of NCOR1 in iYAP5SA MCF10A cells led to the superinduction of cluster 1 genes following YAP5SA induction, whereas cluster 2 genes did not exhibit stronger induction (Fig. 6B,C; Appendix Fig. S3B). Since these experiments identify NCOR1/2 as a JUN-interacting protein critical for its ability to repress YAP target genes, we next tested whether YAP and/or JUN are able to recruit NCOR1/2 to common genomic sites, and whether the binding behavior is consistent with recruitment by JUN in a FOS-independent manner (Fig. 6D–I).

We performed CUT&RUN experiments for JUN, FOS, and NCOR1 in iYAP5SA cells under four conditions: (1) control condition (empty vector, iYAP5SA OFF), (2) iYAP5SA ON, (3) JUN WT overexpression, and (4) JUN M14 overexpression (Fig. 6D–J). As before, iYAP5SA induction led to a strong recruitment (~16-fold) of JUN to joint YAP/JUN sites which was not paralleled by FOS recruitment to these sites (Fig. 6D,G,J). This caused a sharp increase in the JUN/FOS ratio at joint YAP/JUN sites (Fig. 6E,H), which is consistent with a FOS-independent function of JUN. NCOR1, on the other hand, was efficiently recruited by iYAP5SA induction (Fig. 6D,G,J). Overexpression of

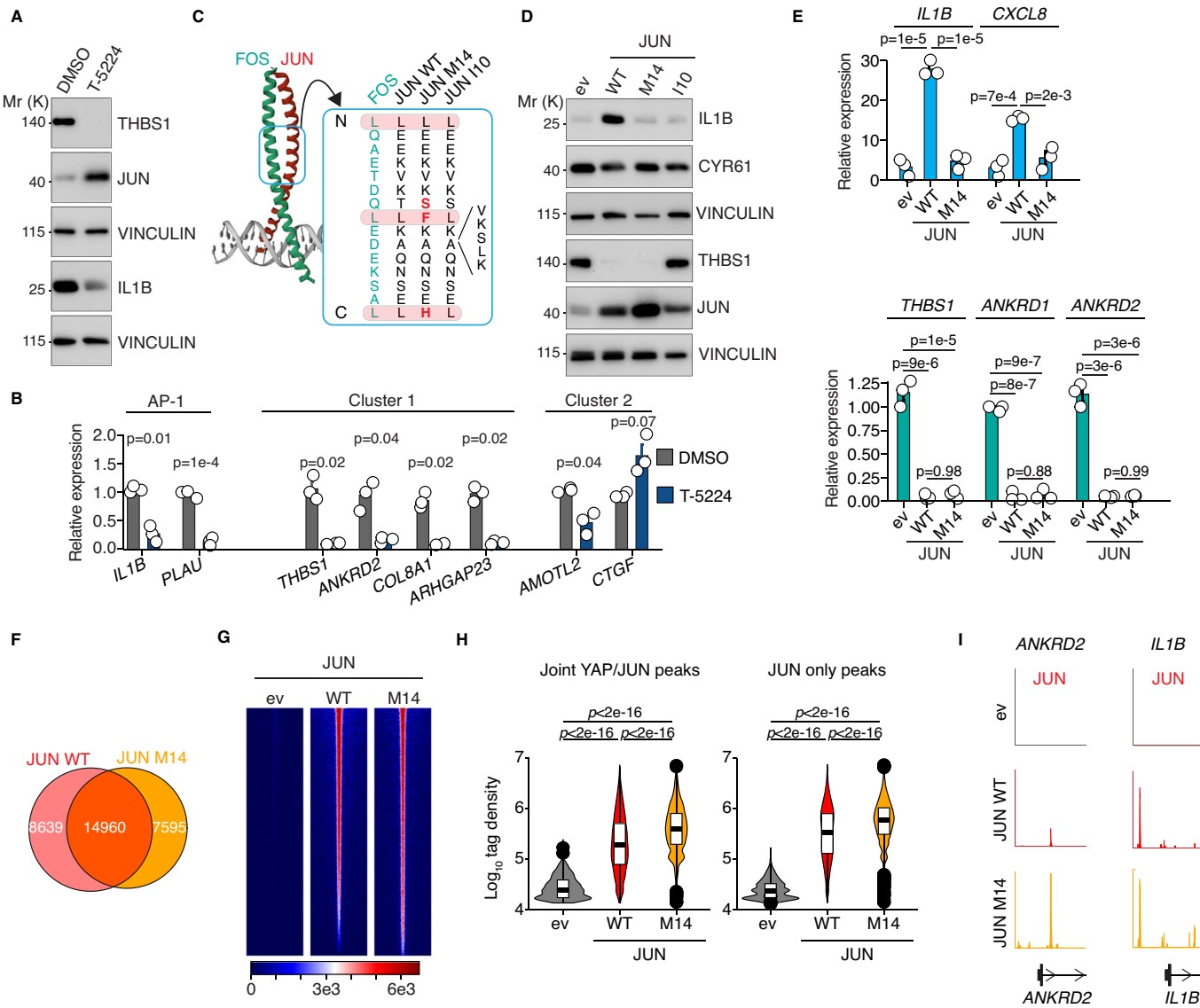

**Figure 4. Canonical AP-1 function and YAP inhibition are distinct JUN properties.**

(A) Immunoblot of MCF10A cells that were treated with 100 µM of the AP-1 inhibitor T-5224 for 20 h. (B) qRT-PCR of MCF10A cells that were treated with 100 µM of the AP-1 inhibitor T-5224 for 20 h. $n = 3$ biological replicates per group. Two-sided Welch test with Benjamini-Hochberg correction. (C) Schematic illustrating the mutations of JUN M14 and JUN I10. A blue box in the JUN::FOS crystal structure (left) marks the localization of the mutation in the JUN leucine zipper. The point mutations are shown in red, and the localization of the four amino acid insertion in JUN I10 is indicated. (D) Immunoblots of MCF10A cells infected with the indicated JUN alleles or an empty vector control (ev). (E) qRT-PCR analysis of MCF10A cells infected with JUN WT, JUN M14 or empty vector control (ev). One-way ANOVA with Tukey HSD post hoc test ($n = 3$, biological replicates). (F) Venn diagram of CUT&RUN peaks of JUN WT and JUN M14. CUT&RUN was performed in MCF10A JUN KO cells that were either reconstituted with JUN WT or JUN M14. (G) CUT&RUN heatmaps for JUN WT and JUN M14 binding to all 31,965 JUN peaks defined previously (see JUN Venn diagram Fig. 3D). (H) Violin plots for the tag densities of JUN WT and JUN M14 CUT&RUN data in 5139 joint YAP/JUN or 18,465 JUN only peaks ($n = 3$, biological replicates). Two-sided Wilcox test with Benjamini-Hochberg correction. (I) Representative CUT&RUN sequencing tracks that illustrate the binding of JUN WT and JUN M14 to *ANKRD2* and *IL1B*. The error bars in this figure indicate the standard error of the mean. All boxplots encompass data points between the first and third quartiles. The median is indicated by a horizontal line. Whiskers extend to 1.5x interquartile range; any data points outside that range are shown as individual dots. Source data are available online for this figure.

JUN WT or JUN M14 alone resulted in a less pronounced recruitment of JUN to the same sites (Fig. 6F,I). Correspondingly, the recruitment of NCOR1 was substantially weaker compared to that achieved by iYAP5SA induction indicating that YAP recruitment is the limiting factor regarding the recruitment of repressive complexes containing JUN and NCOR1 (Fig. 6F,I).

## JUN suppresses YAP-dependent liver cancers

As described before (Fig. 2E), and shown by others (Maglic et al, 2018), YAP can induce JUN expression on protein level. This suggests that JUN is part of a negative feedback loop restraining expression of cluster 1 genes (Fig. 7A). We thus wondered whether

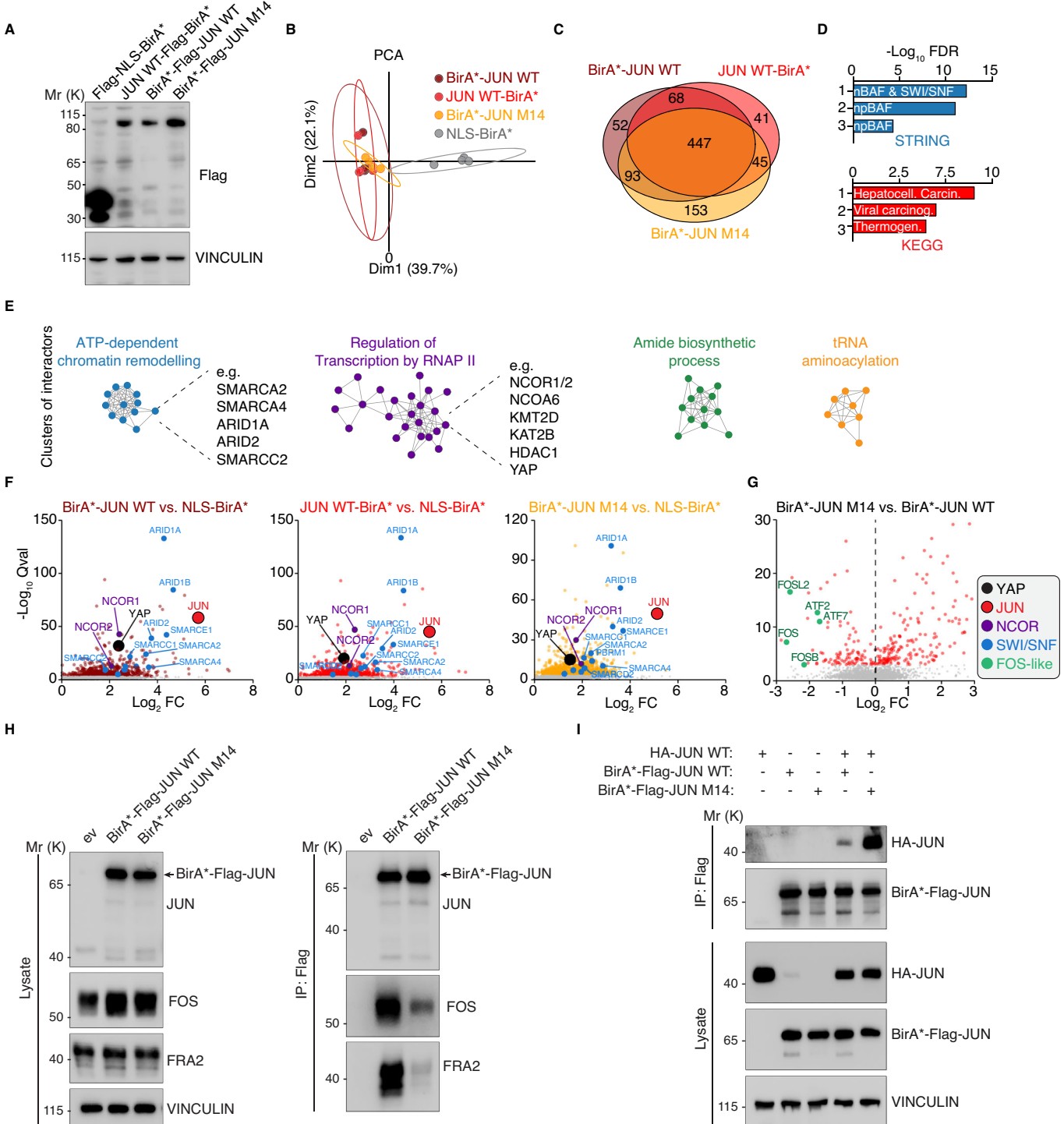

**Figure 5.  JUN M14 shares a similar interactome with JUN WT.**

(**A**) Immunoblot of MCF10A JUN KO cells reconstituted with the indicated BirA*-Flag alleles. (**B**) Principal component analysis (PCA) for the BioID experiment. (**C**) Venn diagram for the high confidence interactors (q-val < 0.001, Log$_2$FC > 1) of the indicated BirA*-Flag JUN alleles identified by BioID. (**D**) Functional annotation of the 447 common JUN interactors using the STRING or KEGG database. FDR false discovery rate. (**E**) Clustering analysis of 447 common JUN interactors. (**F**) Volcano plots for the enrichment of the different BirA* JUN fusion proteins versus the NLS-BirA* control. (**G**) Volcano plots for the enrichment of the BirA*-Flag JUN WT versus BirA*-Flag JUN M14. (**H**) Co-immunoprecipitation experiments from MCF10A JUN KO cells reconstituted with the indicated alleles. BirA* fusion proteins were immunoprecipitated with Flag, and precipitates were assayed for endogenous FOS and FRA2. (**I**) Exogeneous co-immunoprecipitation experiments from 293T cells transfected with the indicated expression constructs. BirA* fusion proteins were immunoprecipitated with Flag, and precipitates were assayed for HA-JUN WT. Source data are available online for this figure.

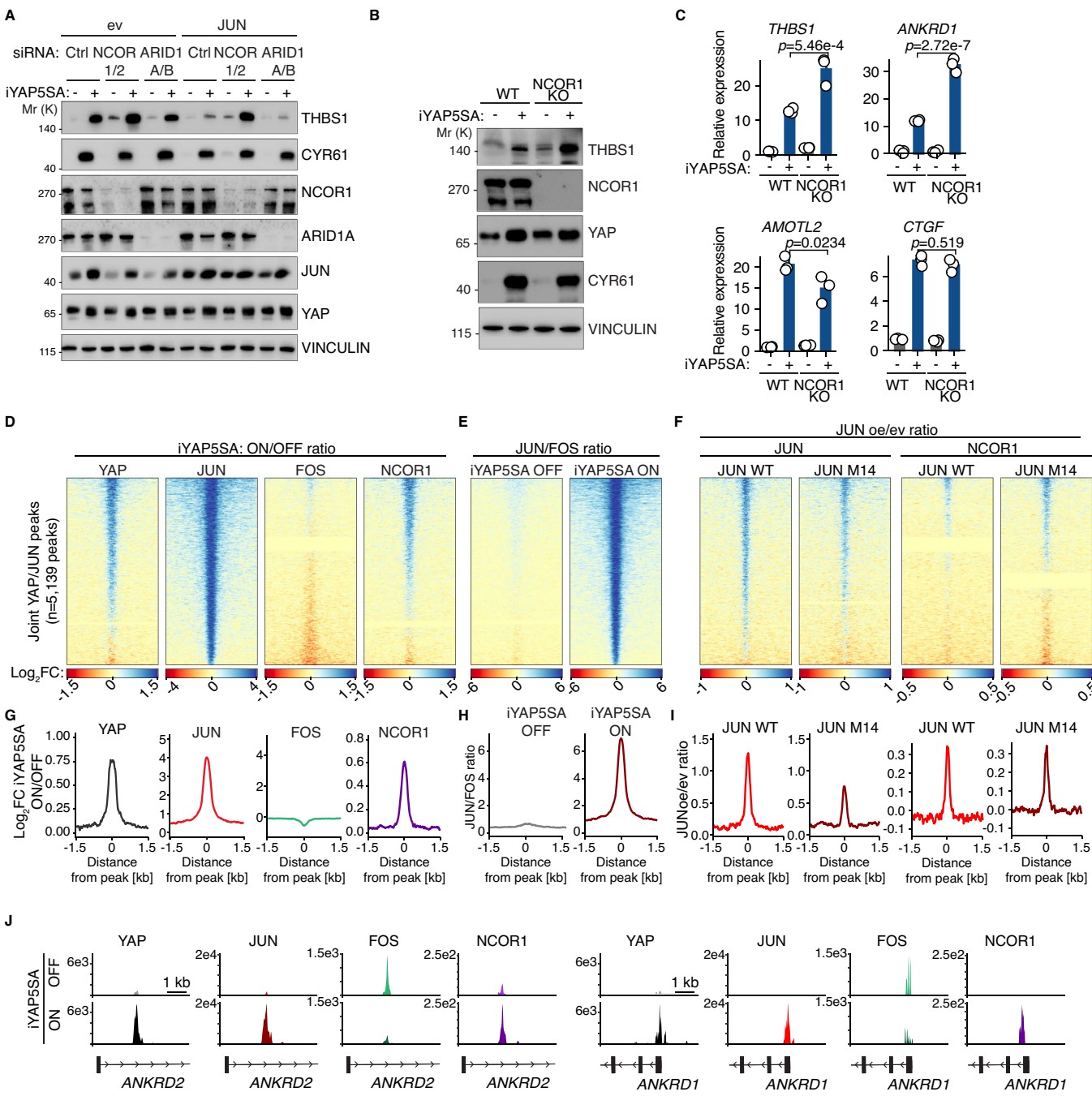

**Figure 6. JUN depends on NCOR1/2 to repress YAP target genes.**

(A) Immunoblot from iYAP5SA MCF10A cells infected with JUN WT or empty vector control (ev) which were transfected with the indicated siRNAs. YAP5SA expression was induced by doxycycline. $n = 2$ (biological replicates). (B) Immunoblot from parental iYAP5SA MCF10A or NCOR1 KO cells in which YAP5SA expression was induced by doxycycline. $n = 2$ (biological replicates). (C) qRT-PCR analysis of the same cells as in (B) ($n = 3$, biological replicates). One-way ANOVA with Tukey HSD post hoc test. (D) CUT&RUN heatmaps from iYAP5SA MCF10A cells for YAP, JUN, FOS, and NCOR1 5335 joint YAP/JUN peaks. The log2 ratio of the YAP5SA ON vs. YAP5SA OFF ratio is plotted here, $n = 2$ (biological replicates). (E) CUT&RUN heatmaps from the same cells as in (D). The log2 ratio of the JUN/FOS signal in the YAP5SA ON vs. YAP5SA OFF condition is plotted here, $n = 2$ (biological replicates). (F) CUT&RUN heatmaps from iYAP5SA MCF10A cells infected with JUN overexpression constructs (JUN WT and JUN M14) or an empty vector control (ev). The log2 ratio of the JUN oe vs. ev ratio for JUN and NCOR1 is plotted here, $n = 2$ (biological replicates). (G–I) Histograms of the mean signal for the heatmaps in (D–F). (J) Representative CUT&RUN sequencing tracks from iYAP5SA MCF10A cells. The error bars in this figure indicate the standard error of the mean. Source data are available online for this figure.

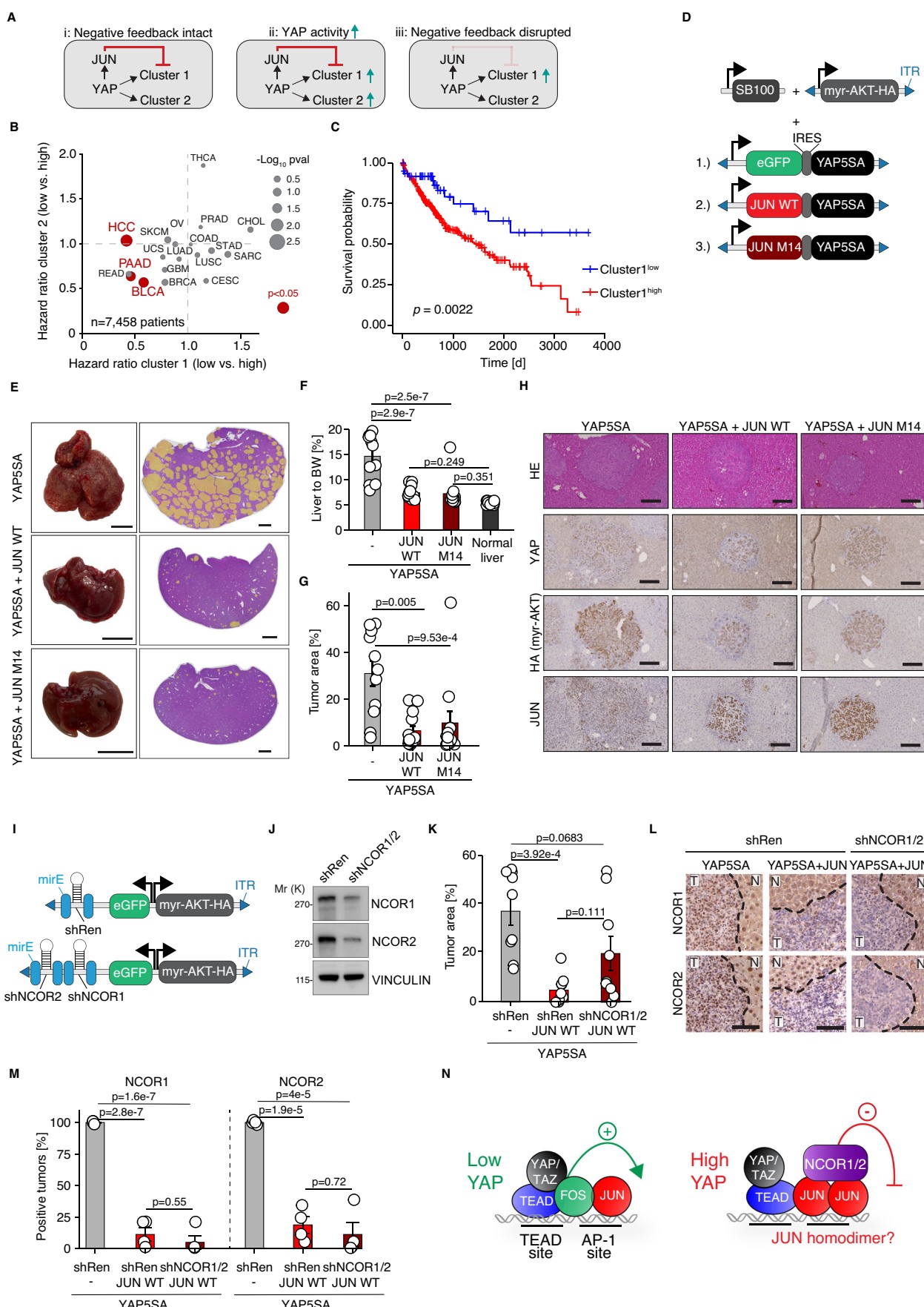

**Figure 7.    JUN represses YAP-dependent liver cancer growth.**

(A) Rationale to identify conditions in which specifically the negative feedback between JUN and YAP is lost. (B) Differential survival analysis in TCGA data sets for 7458 cancer patients in 19 different cancer types. Per cancer type, each patient was analyzed for expression levels of cluster 1 and cluster 2 genes, respectively. Patients were stratified for low and high expression of both clusters, and the survival probability (hazard ratio) per cancer type was plotted against each other in both strata. Cancers with significant survival differences ($p < 0.05$) for cluster 1 genes are highlighted in red. (C) Kaplan Meier curve for HCC patients that were stratified for expression of cluster 1 genes. Log-rank test. (D) Schematic for the liver cancer model by hydrodynamic tail vein injection (HDTVI). SB sleeping beauty transposase, ITR inverted terminal repeat. (E) Representative photos and H&E sections of mouse livers six weeks after HDTVI. Scale bars = 1 cm (left), 2 cm (right). (F) Liver-to-body weight ratios for all mice six weeks after HDTVI. One-way ANOVA with Tukey HSD post hoc test. YAP5SA only ($n = 11$), YAP5SA + JUN WT ($n = 13$), YAP5SA + JUN M14 ($n = 12$), normal liver ($n = 5$). (G) Quantification of tumor areas (tumor area/liver area). One-way ANOVA with Tukey HSD post hoc test. YAP5SA only ($n = 11$), YAP5SA + JUN WT ($n = 13$), YAP5SA + JUN M14 ($n = 12$). (H) Representative immunohistochemical analyses of HDTVI liver tumors. Scale bar = 200 μm. (I) Schematic of the modified myr-AKT-HA vector to co-express mirE shRNAs. shRen shRenilla (non-targeting control). (J) Immunoblots from NIH3T3 cells expressing a shRen control shRNA or a shNCOR1/2 tandem shRNA. Stable integration was achieved using a transposase system, $n = 1$. (K) Quantification of tumor areas (tumor area/ liver area). One-way ANOVA with Tukey HSD post hoc test. YAP5SA + shRen ($n = 9$), YAP5SA + JUN + shRen ($n = 10$), YAP5SA + JUN + shNCOR1/2 ($n = 9$). (L) Representative immunohistochemical analyses for NCOR1 and NCOR2 of HDTVI liver tumors from (J). T tumor, N normal tissue (hepatocytes). Scale bar = 100 μm. (M) Quantification of NCOR1 and NCOR2 signals in tumors for the indicated groups. Per mouse, at least 20 tumors were randomly selected and the signal was quantified as positive or negative. $n = 4$ per experimental group. One-way ANOVA with Tukey HSD post hoc test. (N) Model proposing how JUN restrains supraphysiological YAP levels. Biological replicates were used for all experiments of this figure. The error bars in this figure indicate the standard error of the mean. Source data are available online for this figure.

this negative feedback loop gets disrupted in cancer to unleash YAP's full transcriptional potential. We performed a differential analysis using genes from cluster 1 and cluster 2 as readout of YAP transcriptional activity to identify scenarios in which specifically this feedback loop is disrupted (Fig. 7A,B). That way, one should be able to discriminate between a general increase in YAP activity (e.g., by YAP overexpression), and a potential disruption of the feedback loop since the latter would specifically affect cluster 1 genes. To this end, 19 different cancer entities with 7458 patients in TCGA data sets were analyzed for a specific survival benefit when comparing expression of cluster 1 vs. cluster 2 genes (Fig. 7B). In this analysis, hepatocellular carcinoma (HCC) stood out, as in particular the survival probability of patients with low expression of cluster 1 genes was strongly increased, whereas the hazard ratio for cluster 2 genes was largely unaffected (Fig. 7B,C). To test whether JUN could suppress YAP-dependent liver cancer, we performed hydrodynamic tail vein injection (HDTVI) in conjunction with a sleeping beauty-based approach to stably express genes in the livers of C57BL/6J wild-type mice. YAP overexpression in combination with constitutively active myristoylated AKT (myr-AKT) leads to induction of hepatocellular carcinomas only a few weeks after HDTVI (Yamamoto et al, 2017). To investigate how JUN affects the oncogenic potential of YAP in this context, we co-overexpressed myr-AKT with the following constructs (Fig. 7D): GFP-IRES-YAP5SA (Ctrl), JUN WT-IRES-YAP5SA (JUN WT), and JUN M14-IRES-YAP5SA (JUN M14). Six weeks after HDTVI, the mice were sacrificed, and the livers were analyzed (Fig. 7E–H). The livers of the YAP5SA condition were strongly enlarged and showed numerous macroscopically visible tumor nodules whereas the JUN WT, as well as the JUN M14, livers appeared largely normal (Fig. 7E). The liver to body weight ratios of YAP5SA-JUN WT and YAP5SA-JUN M14 animals were comparable to wild-type mice, while it was significantly elevated in YAP5SA animals (Fig. 7F). YAP5SA livers showed a multifocal tumor growth with large tumor lesions whereas those lesions were barely detectable in YAP5SA-JUN WT/M14 livers, demonstrating that JUN's repressive effect on YAP activity potently interferes with liver tumor growth (Fig. 7E–G). Notably, JUN here reduced tumor growth despite its ability to block apoptosis (Fig. EV5A,B) as described before in JUN-deficient livers (Eferl et al, 2003). On sections, all tumors were positive for HA-myr-AKT, nuclear YAP, and YAP5SA-JUN WT/

M14 tumors showed strong JUN expression, demonstrating that the proteins were stably expressed as expected (Fig. 7H). To test whether JUN's ability to restrain tumor growth depends on NCOR1/2 expression, we cloned mirE-based (Fellmann et al, 2013) tandem shRNA constructs targeting NCOR1 and NCOR2 into the myr-AKT-HA vector (Fig. 7I). The NCOR1/2 shRNAs were not very potent since they reduced protein expression only by 50-60% compared to the shRen non-targeting control (Fig. 7J). Next, we conducted HDTVI experiments as before, here using three experimental groups: (1) YAP5SA+shRen, (2) YAP5SA + JUN +shRen, and (3) YAP5SA + JUN+shNCOR1/2. Despite the weak knockdown efficiency, NCOR1/2 depletion restored tumor growth in several animals even though it did not reach significance when comparing all animals in the cohort (Fig. 7K). When analyzing the tumors of this cohort for NCOR1/2 expression, we noticed that a significant proportion of YAP5SA + JUN tumors had lost NCOR1/2 expression even in the shRen control where no shRNA targeting NCOR1/2 was expressed (Fig. 7L,M). Since this phenomenon did not manifest in the YAP5SA+shRen group (Fig. 7L,M), it suggests that there is selective pressure on evolving tumors to diminish or eliminate NCOR1/2 expression under conditions of high JUN expression. Hence, the depletion of NCOR1/2 by a relatively weak shRNA has only a marginal additive effect. In summary, this unexpected but unbiased finding reinforces the association between NCOR1/2 and JUN's capacity to suppress YAP target genes and demonstrates a role in YAP-dependent liver cancer.

## Discussion

Numerous ChIP-Seq studies in cancer cells have documented a substantial overlap of YAP/TAZ, TEAD, and AP-1 binding in the genome (Stein et al, 2015; Zanconato et al, 2015), and JUN/FOS AP-1 complexes are required for full transcriptional activity of YAP/TAZ (Shao et al, 2014). Our work now identifies an unexpected FOS-independent role of JUN whose function is to buffer unrestrained supraphysiological YAP/TAZ activity on chromatin level at a large fraction of YAP/TAZ target genes. Based on our CUT&RUN data (Fig. 3), we propose that JUN acts as a rheostat to limit these supraphysiological—potentially oncogenic—YAP levels specifically at weak enhancers that become active only

when a certain YAP activity is exceeded, leading to enhancer invasion of YAP at these sites.

Most likely, JUN together with its associated corepressor complexes, gets recruited by protein-protein interactions between YAP and JUN to weak enhancer. These interactions are further stabilized by DNA-binding of JUN to AP-1 DNA sites (Fig. 7N). These sites seem to be pre-occupied by activating JUN::FOS heterodimers, so that JUN recruitment defines an AP-1 switch: from activating JUN::FOS to repressive JUN complexes, potentially JUN::JUN homodimers. Currently, it is unclear how the regulation of this switch is orchestrated: it cannot be simply explained by different ratios of JUN and FOS since JUN overexpression does not lead to a general switch of AP-1 to a repressor because it still induces canonical AP-1 targets, such as *IL1B*. It is thus conceivable that YAP locally recruits additional proteins that mediate this switch, but this needs to be investigated in the future.

Hepatocellular carcinoma is certainly one of the best characterized tumor entities with respect to the effects of uncontrolled YAP/TAZ activity. We now identify JUN as a component of a negative feedback mechanism with tumor suppressive properties to protect the organism from oncogenic YAP levels. At first glance, this finding seems counter-intuitive since deletion of JUN leads to increased survival in a chemically induced liver cancer model (Eferl et al, 2003) which would suggest pro-oncogenic properties of JUN in liver cancer. However, deletion of JUN leads to removal both both JUN activities: activating as well as repressive functions. It is therefore most likely context-dependent which of the two functions is more important for carcinogenesis, and this is certainly influenced by the driver mutations in each tumor. This dichotomy nonetheless implies that some of the described JUN knockout phenotypes may need to be reevaluated in this context, e.g., using JUN mutants such as the JUN M14 mutant that can uncouple the two JUN functions.

The uncoupling of JUN's dual functions can also be achieved pharmacologically, for instance, with T-5224, which disrupts activating AP-1 functions (Fig. 4B). Notably, T-5224 is highly efficient in reducing liver tumors driven by YAP (Koo et al, 2020).

In conclusion, our work defines a new layer of YAP/TAZ regulation in the context of the intricate AP-1 network which could be utilized for developing novel cancer therapies.

# Methods

### Reagents and tools table

| Antibodies | | | |
|---|---|---|---|
| **Name** | **Source** | **Identifier** | **Application, concentration** |
| ARID1A/BAF250A(D2A8U) | Cell Signaling | 12354 | WB, 1:1000 |
| FOS (i.e., c-FOS) | Abcam | ab222699 | WB, 1:1000 |
| FOS (i.e., c-FOS) | Cell Signaling | 2250 | CUT&RUN: 1:100 |
| JUN (c-JUN, 60A8) | Cell Signaling | 9165 | WB, 1:1000 CUT&RUN, 1:100 IHC, 1:100 (TE) |
| CYR61 (D4H5D) | Cell Signaling | 14479 | WB, 1:1000 |
| FRA2 (D2F1E) | Cell Signaling | 19967 | WB, 1:1000 |
| HA Epitope Tag | Rockland | 600-401-384 | WB, 1:500 IHC, 1:2000 (citrate) |
| Flag M2-tag | Sigma-Aldrich | F1804 | WB. 1:1000 |
| IL-1β (D3U3E) | Cell Signaling | 12703 | WB, 1:1000 |
| IRS-1 (59G8) | Cell Signaling | 2390 | WB, 1:1000 |
| IgG | Sigma-Aldrich | I5006 | CUT&RUN, 1:100 |
| NCOR1 | Cell Signaling | 5948 | WB: 1:1000 CUT&RUN, 1:100 IHC: 1:100 (TE) |
| NCOR1 (E4S4N) | Cell Signaling | 34271 | WB, 1:1000 |
| NCOR2 | Abcam | ab5802 | WB: 1:2000 IHC: 1:100 (TE) |
| p-YAP (Ser127) | Cell Signaling | 4911 | WB: 1:1000 |
| TEF-1 (TEAD1) | BD Transduction Laboratories | 610923 | CUT&RUN, 1:100 |
| Thrombospondin1 (THBS1, A6.1) | Santa Cruz | sc-59887 | WB, 1:500 |
| V5-Tag | Cell Signaling | 13202 | WB, 1:1000 |
| VINCULIN | Sigma-Aldrich | V9131 | WB, 1:10,000 |
| YAP (D8H1X) | Cell Signaling | 14074 | WB, 1:2000 IHC, 1:200 (citrate) |
| YAP1 | Abcam | ab52771 | CUT&RUN, 1:100 |
| Cleaved Caspase3 | Cell Signaling | 9661 | IHC 1:200 (citrate) |
| H3K27ac | Millipore | MABE647 | CUT&RUN, 1:100 |
| H3K4me1 (D1A9) | Cell Signaling | 5326 | CUT&RUN, |
| Goat Anti-Rabbit Immunoglobulins/HRP | Agilent | P044801-2 | WB, 1:5000 IHC, 1:1000 |
| Goat Anti-Mouse Immunoglobulins/HRP | Agilent | P044701-2 | WB, 1:5000 IHC, 1:1000 |

| siRNAs | | |
|---|---|---|
| **Gene target** | **Dharmacon ID** | **Specification** |
| NTC | D-001810-10 | ON-TARGETplus Non-targeting Control Pool |
| ARID1A | L-017263-00 | ON-TARGETplus Human ARID1A (8289) siRNA - SMARTpool |
| ARID1B | L-013970-01 | ON-TARGETplus Human ARID1B (57492) siRNA - SMARTpool |
| NCOR1 | L-003518-00 | ON-TARGETplus Human NCOR1 (9611) siRNA - SMARTpool |
| NCOR2 | L-020145-01 | ON-TARGETplus Human NCOR2 (9612) siRNA - SMARTpool |

| Oligonucleotides to amplify sgSAM cassette for NGS library preparation | | |
|---|---|---|
| **Primer** | **Sequence 5′–3′** | |
| | **Forward** | **Reverse** |
| SAM pre-amplification | GGCCTATTTCCCATGATTCCT | GCCAATTCCCACTCCTTTCA |
| SAM#1 | AATGATACGGCGACCACCGAGATCTACACTCTTTCCCTACACGACGCTCTTCCGATCTGATCGCTTTATATATCTTGTGGAAAGGACGAAACAC*C | CAAGCAGAAGACGGCATACGAGATAGCTTCAGGTGACTGGAGTTCAGACGTGTGCTCTTCCGATCTGCCAAGTTGATAACGGACTAGCCT*T |
| SAM#2 | AATGATACGGCGACCACCGAGATCTACACTCTTTCCCTACACGACGCTCTTCCGATCTATCGCTTTATATATCTTGTGGAAAGGACGAAACAC*C | CAAGCAGAAGACGGCATACGAGATGCGCATTAGTGACTGGAGTTCAGACGTGTGCTCTTCCGATCTGCCAAGTTGATAACGGACTAGCCT*T |
| SAM#3 | AATGATACGGCGACCACCGAGATCTACACTCTTTCCCTACACGACGCTCTTCCGATCTTCGCTTTATATATCTTGTGGAAAGGACGAAACAC*C | CAAGCAGAAGACGGCATACGAGATCATAGCCGGTGACTGGAGTTCAGACGTGTGCTCTTCCGATCTGCCAAGTTGATAACGGACTAGCCT*T |
| SAM#4 | AATGATACGGCGACCACCGAGATCTACACTCTTTCCCTACACGACGCTCTTCCGATCTCGCTTTATATATCTTGTGGAAAGGACGAAACAC*C | CAAGCAGAAGACGGCATACGAGATTTCGCGGAGTGACTGGAGTTCAGACGTGTGCTCTTCCGATCTGCCAAGTTGATAACGGACTAGCCT*T |
| SAM#5 | AATGATACGGCGACCACCGAGATCTACACTCTTTCCCTACACGACGCTCTTCCGATCTTGATGCTTATATATCTTGTGTGGAAAGGACGAAACAC*C | CAAGCAGAAGACGGCATACGAGATGCGCGAGAGTGACTGGAGTTCAGACGTGTGCTCTTCCGATCTGCCAAGTTGATAACGGACTAGCCT*T |

| SAM#6 | AATGATACGGCGACCA CCGAGATCTACACTCTT TCCCTACACGACGCTCT TCCGATCTGTGATCGCT TTATATATCTTGTGGAA AGGACGAAACAC*C | CAAGCAGAAGACGGCA TACGAGATCTATCGCTG TGACTGGAGTTCAGACG TGTGCTCTTCCGATCTGC CAAGTTGATAACGGACT AGCCT*T |
|---|---|---|

### qRT-PCR oligonucleotides

| Primer | Sequence 5′–3′ | |
|---|---|---|
| | Forward | Reverse |
| ANKRD1 | AGTAGAGGAACTGGTCACTGG | TGTTTCTCGCTTTTCCACTGTT |
| ANKRD2 | GCACAGGAGGAGGAGAATGA | CTCTTGGCCCTTCACCTTCT |
| ARHGAP23 | GGCTGGTAAAGGTGAATGGGG | TAGGCATCCTGGGAGTAGGC |
| AMOTL2 | TCAGGAGATGGAAAGCAGGTT | GAAAACAGATGGCACCGACTT |
| COL8A1 | GGGAGTGCTGCTTACCATTTC | AGCGGCTTGATCCCATAGTAG |
| CTGF | CAGCATGGACGTTCGTCTG | AACCACGGTTTGGTCCTTGG |
| CYR61 | CTTGTTGGCGTCTTCGTCG | AGCCTGGTCAAGTGGAGAAG |
| CXCL8 | TTTTGCCAAGGAGTGCTAAAGA | AACCCTCTGCACCCAGTTTTC |
| IL1B | ATGATGGCTTATTACAGTGGCAA | GTCGGAGATTCGTAGCTGGA |
| PLAU | GGCTTAACTCCAACACGCAA | TATACATCGAGGGCAGGCAG |
| THBS1 | GATGTGGAAGCAAGTCACCC | CTTTCACAGAAAGGCCCGAG |
| B2M | GTGCTCGCGCTACTCTCTC | GTCAACTTCAATGTCGGAT |

### shRNA cloning

| 97mer/oligo | Sequence 5′–3′ |
|---|---|
| shRen | TGCTGTTGACAGTGAGCGCAGGTGCCAAGAAGTTTCCTAATAGTGAAGCCACAGATGTATTAGGAAACTTCTTGGCACCTTTGCCTACTGCCTCGGA |
| shNCOR1.1 | TGCTGTTGACAGTGAGCGCAAAGACTGAATTTTAAACTAATAGTGAAGCCACAGATGTATTAGTTTAAAATTCAGTCTTTATGCCTACTGCCTCGGA |
| shNCOR1.2 | TGCTGTTGACAGTGAGCGATCCGCATCAAGTGATAACTAATAGTGAAGCCACAGATGTATTAGTTATCACTTGATGCGGAGTGCCTACTGCCTCGGA |
| shNCOR2.1 | TGCTGTTGACAGTGAGCGCCCTGACCAAGAAGAATGAAAATAGTGAAGCCACAGATGTATTTTCATTCTTCTTGGTCAGGTTGCCTACTGCCTCGGA |
| shNCOR2.2 | TGCTGTTGACAGTGAGCGCACACATGTTGTTCCAATTAGATAGTGAAGCCACAGATGTATCTAATTGGAACAACATGTGTATGCCTACTGCCTCGGA |
| mirE fwd | TGAACTCGAGAAGGTATATTGCTGTTGACAGTGAGCG |
| mirE rev | TCTCGAATTCTAGCCCCTTGAAGTCCGAGGCAGTAGGC |

### Commercial assays

| Name | | Source | Identifier |
|---|---|---|---|
| innuMix qPCR DSGreen Standard | | Analytik Jena | 845-AS-1300200 |
| NEBNext® Ultra RNA Library Prep kit for Illumina | | NEB | E7530 |
| NEBNext® Poly(A) mRNA Magnetic Isolation Module | | NEB | E7490 |
| NEBNext® Multiplex Oligos for Illumina (Dual index kit) | | NEB | E7600 |
| NEBNext® Ultra™ II DNA Library Prep Kit for Illumina | | NEB | E7645 |
| Lexogen's QuantSeq™ 3′ mRNA-Seq Kit | | Lexogen | 015 |
| QIAamp DNA Blood Maxi Kit | | Qiagen | 51194 |
| RNeasy® Micro Kit | | Qiagen | 74004 |
| QIAGEN Plasmid Maxi Kit | | Qiagen | 12165 |
| Endotoxin-free plasmid DNA purification kit (NucleoBond PC 500 EF) | | Macherey-Nagel | 740550 |

### Chemicals, peptides and recombinant proteins

| Name | Source | Identifier |
|---|---|---|
| Anti-Flag® M2 Affinity Gel | Sigma-Aldrich | A2220-5ML |
| B27 supplement | Thermo Fisher Scientific | 17504044 |
| DMEM/F-12 | Thermo Fisher Scientific | 31330095 |
| DMEM (GlutaMAX™) | Thermo Fisher Scientific | 61965059 |
| Doxycycline | Sigma-Aldrich | D9891 |
| Digitonin | Sigma-Aldrich | D141-500 MG |
| Fetal Bovine Serum (FBS) | Gibco | F7524-500ML |
| Insulin solution human | Sigma-Aldrich | I9278 |
| Human recombinant EGF | Biomol | 50349.1000 |
| Cholera toxin | Sigma-Aldrich | C8052-2MG |
| Hydrocortizone | Sigma-Aldrich | H0888-5G |
| Heparin | Sigma-Aldrich | H3149-50KU |
| Horse serum | Thermo Fisher Scientific | 16050122 |
| Penicillin/Streptomycin | Thermo Fisher Scientific | 15140122 |
| Opti-MEM® | Thermo Fisher Scientific | 31985047 |
| Lipofectamine RNAiMAX | Thermo Fisher Scientific | 13778150 |
| Lipofectamine 3000 | Thermo Fisher Scientific | L3000015 |
| T-5224 | MedChemExpress | HY-12270 |
| TRULI | MedChemExpress | HY-138489 |
| Protamine sulfate | Sigma-Aldrich | P4505-1G |
| PEI-Max | Polyscience | 24765-1 |
| peqGOLD TriFast™ | VWR | 30-2010 |
| protease inhibitor cocktail | Sigma-Aldrich | P8340-5ML |
| M-MLV Reverse Transcriptase | Promega | M1705 |
| Methylcellulose | Sigma-Aldrich | M7027-100g |
| FLAG® Peptide | Sigma-Aldrich | F3290 |
| indole-3-acetic acid | Sigma-Aldrich | I3750-5G-A |
| iodoacetamide | Sigma-Aldrich | I1149-5G |
| 4-Thiouridine | Sigma-Aldrich | T4509-25MG |
| random hexamer primers | Sigma-Aldrich | 11034731001-2MG |
| ROX Reference Dye | Thermo Fisher Scientific | 12223012 |
| RNaseA | Roth | 7156.1 |
| proteinase K | Roth | 7528.1 |
| Vector® ImmPACT DAB Peroxidase (HRP)Sub. | Vector Laboratories | SK-4105 |
| Streptavidin Sepharose High Performance Beads | Cytiva | 17-5113-01 |
| BioMag.Plus Concanavalin A magnetic beads | Polyscience | 86057-10 |
| Spermidine | Sigma-Aldrich | S0266-5G |
| OmniPur® DTT | Millipore | 3860-OP |
| GlycoBlue | Thermo Fisher Scientific | AM9516 |
| Clarity Western ECL Substrate | Bio-Rad | 1705061 |

### Plasmid constructs

| | Backbone | Insert |
|---|---|---|
| **Lentivirus packaging** | | |
| psPAX2 | | Lentiviral packaging plasmid |
| pMD2.G | pMD2.G | VSV-G envelope expressing plasmid |
| **Lentiviral constructs** | | |
| pInducer-21-Strep-YAP5SA | pInducer-21 | Doxycycline-inducible YAP5SA |
| LeGO-iG2-Puro-Flag-YAP5SA | LeGO-iG2-Puro-SFFV | Flag-YAP 5SA |
| LeGO iG2-Puro-Flag-YAP5SA S94A | LeGO-iG2-Puro SFFV | Flag-YAP 5SA S94A |
| LeGO-iG2-Puro-EF1a-JUN WT | LeGO-iG2-Puro-EF1a | JUN WT |

| | | |
|---|---|---|
| LeGO-iG2-Puro-EF1a-HA-JUN WT | LeGO-iG2-Puro-EF1a | HA-JUN WT |
| LeGo-iG2-Puro-HA-JUN | LeGO-iG2-Puro SFFV | HA-JUN WT |
| LeGO-iG2-Puro-HA-JUNB | LeGO-iG2-Puro SFFV | HA-JUNB |
| LeGO-iG2-Puro-JUND-HA | LeGO-iG2-Puro SFFV | JUND-HA |
| LeGO-iG2-Puro-EF1a-JUN M14 | LeGO-iG2-Puro-EF1a | JUN M14 |
| LeGO-iG2-Puro-EF1a-JUN I10 | LeGO-iG2-Puro-EF1a | JUN I10 |
| LeGO-iG2-Puro-EF1a-JUN-V5-AID | LeGO-iG2-Puro-EF1a | JUN-V5-AID |
| LeGO-iG2-Puro-NLS-BirA-Flag* | LeGO-iG2-Puro | NLS-BirA-Flag* |
| LeGO-iG2-EF1a-BirA*-Flag-JUN WT | LeGO-iG2-Puro-EF1a | BirA*-Flag-JUN WT |
| LeGO-iG2-EF1a-JUN WT-BirA*-Flag | LeGO-iG2-Puro-EF1a | JUN WT-BirA*-Flag |
| LeGO-iG2-EF1a-BirA*-Flag-JUN M14 | LeGO-iG2-Puro-EF1a | BirA*-Flag-JUN M14 |
| pRRL-TIR1-Hygro | pRRL-Hygro | TIR1 |
| SGEP | SGEP | shRen |
| **CRISPR plasmid** | | |
| pX461-CAS9-WT-sgJUN | pX461-CAS9-WT | sgJUN |
| pMSCV-U6sgRNA(BbsI)-PGKpuro2ABFP-sgNCOR1 | pMSCV-U6sgRNA(BbsI)-PGKpuro2ABFP | sgNCOR1 |
| **SAM plasmids** | | |
| dCas9-VP64 | | |
| MCP-p65-HSF1 | | |
| CRISPR/Cas9 Synergistic Activation Mediator (SAM) sgRNA library | | SAM sgRNA library |
| lenti sgSAM MYC #1 (MS2) zeo | sgRNA (MS2) zeo | sgSAM MYC#1 |
| lenti sgSAM MYC #2 (MS2) zeo | sgRNA (MS2) zeo | sgSAM MYC#2 |
| lenti sgSAM JUN #1 (MS2) zeo | sgRNA (MS2) zeo | sgSAM JUN #1 |
| lenti sgSAM JUN #2 (MS2) zeo | sgRNA (MS2) zeo | sgSAM JUN #2 |
| lenti sgSAM NTC#1(MS2) zeo | sgRNA (MS2) zeo | sgSAM NTC#1 |
| lenti sgSAM NTC#2 (MS2) zeo | sgRNA (MS2) zeo | sgSAM NTC#2 |
| **HDTVI plasmids** | | |
| pCMV(CAT)T7-SB100 | pCMV | SB100 |
| pSBbi-w/oPuro-Myr-AKT-HA | pSBbi-w/oPuro | Myr-AKT-HA |
| pSBbi-w/oPuro-eGFP-IRES-V5-YAP5SA | pSBbi-w/oPuro | eGFP-IRES-V5-YAP5SA |
| pSBbi-w/oPuro-cJUN-WT-IRES-V5-YAP5SA | pSBbi-w/oPuro | cJUN-WT-IRES-V5-YAP5SA |
| pSBbi-w/oPuro-cJUN-M14-IRES-V5-YAP5SA | pSBbi-w/oPuro | cJUN-M14-IRES-V5-YAP5SA |
| pSBbi-w/oPuro-Myr-AKT-HA-shRen | pSBbi-w/oPuro | Myr-AKT-HA eGFP-5'mirE-shRen-3'' mirE- |
| pSBbi-w/oPuro-Myr-AKT-HA-shNCOR1/2 | pSBbi-w/oPuro | Myr-AKT-HA eGFP-5'mirE-shNCOR1-3'mirE-5'mirE-shNCOR2-3'mirE |

## Reagents

A list of reagents (antibodies, plasmids, oligonucleotides, siRNAs, chemicals, commercial kits) is provided in Reagents and Tools.

## Mice

Animal experiments were conducted in accordance with the guidelines and regulations of the state government of Thuringia under the animal experiment license FLI-21-015. Male C57BL/6JRj mice for HDTVI experiments were obtained from Janvier Labs at 5 weeks of age. The mice were kept in individually ventilated cages

(IVCs) under Specific Pathogen Free (SPF) conditions with a 12 h dark/12 h light cycles at 20 °C and 55% relative humidity according to the directives of the 2010/63/EU and GV SOLAS.

## Mammalian cell culture

293T/LentiX cells (Takara Bio, # 632180) and NIH3T3 cells were cultured in DMEM (+GlutaMAX, Thermo Fisher Scientific) supplemented with 10% FBS (Thermo Fisher Scientific) and 1% penicillin-streptomycin (Sigma-Aldrich). MCF10A cells (kind gift from Martin Eilers, University of Würzburg, Germany) were cultured in DMEM/F12 (Thermo Fisher Scientific) supplemented with 5% horse serum (Thermo Fisher Scientific), 1% penicillin-streptomycin (Sigma-Aldrich), 10 µg/ml human insulin (Sigma-Aldrich), 0.5 µg/ml hydrocortisone (Sigma-Aldrich), 0.1 µg/ml cholera toxin (Sigma-Aldrich) and 20 ng/ml human EGF (Biomol). All cells were kept at 5% $CO_2$, 95% relative humidity and 37 °C and were regularly tested for mycoplasma contamination by PCR.

Transient transfections were carried out using polyethylenei-mine (PEI-Max, Sigma-Aldrich) with Opti-MEM reduced serum medium (Thermo Fisher Scientific).

For siRNA transfections, the Lipofectamine RNAiMAX reagent (Thermo Fisher Scientific) was used. siRNAs were purchased from Dharmacon and are listed in Reagents and Tools.

Expression of inducible YAP5SA mutant in MCF10A cells was performed by incubating the cells with 100 ng/ml doxycycline for 16 h prior to analysis. As control, the cells were treated with the same volume of ethanol.

For T-5224 treatment, the cells were incubated with 100 µM T-5224 (MedChemExpress) for 20 h before processing. Controls were treated with the same volume of DMSO.

For LATS1/2 inhibition, the cells were incubated with 10 µM TRULI (MedChemExpress) for 24 h before processing. Controls were treated with the same volume of DMSO.

## Growth curve

For growth curve experiments, cells were plated at a density of $1.56 \times 10^4$ cells/cm² in 96-well plates. Where appropriate, expression of inducible YAP5SA mutant was induced 12 h after seeding by incubating the cells with 1 µg/ml doxycycline. As control, the cells were treated with the same volume of ethanol.

Live cell imaging was performed using the ®SX5 microscope (Sartorius) during several days with an interval of 3 h. Per well, four images at distinct positions were captured with the 10× objective. For each cell line and/or treatment three biological replicates were analyzed.

## CRISPR-Cas9 KO

To generate MCF10A JUN knockout cells, a small guide RNA targeting the JUN exon (TCGTTCCTCCCGTGAGAG) was cloned into pX461 (a gift from Feng Zhang, Addgene #48140) in which the Cas9 nickase allele was substituted for the wild-type Cas9 allele. MCF10A cells were transfected using Lipofectamine 3000 (Thermo Fisher Scientific). After 48 h, GFP+ cells were isolated by flow cytometry and single cells were seeded in 96-well plates. JUN knockout clones were identified by Western blot. For knockout verification, genomic DNA was isolated and the regions flanking the sgRNA target site were amplified by PCR. The amplicons were cloned into pJET (Thermo Fisher Scientific) and

analyzed by Sanger sequencing, which revealed frame shift mutations leading to generation of premature stop codons.

To generate MCF10A-iYAP5SA NCOR1 knockout cells, a small guide RNA targeting *NCOR1* (GGTGATCCGGCCCTTACGG) was cloned into pMSCV-U6sgRNA(BbsI)-PGKpuro2ABFP (a gift from Sarah Teichmann, Addgene #102796). MCF10A-iYAP5SA cells were transfected using Lipofectamine 3000 (Thermo Fisher Scientific). Single knockout clones were identified and verified same way as described for JUN knockout cells.

## Acute JUN depletion via an auxin-inducible degron

JUN KO MCF10A cells were reconstituted with an auxin-tagged JUN allele (JUN-AID-V5). The cells were infected with a TIR1 overexpression construct, and JUN degradation was induced by adding 100 µM Indole-3-acetic acid (IAA, Sigma-Aldrich).

## Lentiviral transduction

For lentivirus production, LentiX cells were co-transfected with 10 µg psPAX2, 2.5 µg pMD2.G (gift from Didier Trono, Addgene # 12260 and Addgene # 12259) and 10 µg lentiviral vector using PEI-Max (Sigma-Aldrich). Virus-containing supernatants were collected 24 h, 48 h, and 72 h post transfection and pooled. MCF10A cells were infected for 24 h using filtered viral supernatant diluted with culture medium and supplemented with 8 µg/µl protamine sulfate (Sigma-Aldrich). Selection of infected cells with antibiotics was performed 48 h after the infection.

## SAM screening

The genome-wide human CRISPR/Cas9 Synergistic Activation Mediator (SAM) sgRNA library (gift from Feng Zhang, Addgene #1000000057) was amplified as recommended, and balanced sgRNA library distribution was verified by NGS. Clonal MCF10A cells stably expressing SAM components (MCF10A-SAM cells), dCas9-VP64 and MCP-p65-HSF1 (gift from Feng Zhang, Addgene #61425 and #61426), were infected with the lentiviral SAM sgRNA library with a low titer of MOI = 0.5. A library coverage of at least 500-fold was maintained throughout the experiment.

To screen for suppressors of YAP5SA activity, the MCF10A-SAM cells expressing the SAM library were superinfected with YAP5SA and cultured for two weeks to allow outgrowth of cells expressing suppressors of YAP5SA. gDNA was isolated using the QIAamp DNA Blood Maxi Kit (Qiagen). Sequencing libraries were generated using nested PCR. In the first PCR reaction, the integrated sgRNA cassette was amplified and then the second PCR was performed to add Illumina adapters and barcodes for NGS (primer sequences are listed in Reagents and Tools). Libraries were quantified with the Agilent 2100 Bioanalyzer automated electrophoresis system (Agilent Technologies) and subjected to 75 bp single-end Illumina Sequencing on a NextSeq500. Reads were extracted in FastQ format using bcl2fastq v1.8.4 (Illumina).

## SAM analysis

Quality-filtering (>Q30) of the sequencing data and adapter removal was performed with Cutadapt (v2.7). The filtered reads were mapped to a custom reference containing all sgRNA sequences in the SAM library using Bowtie2. To all samples, a pseudocount of +1 was added to avoid division by 0. The reads were normalized by sequencing depth (reads per sgRNA/million mapped reads) and subsequently used in RSA analysis. For the RSA analysis, the following parameters were used, based on the median and the standard deviations (SDs) of the reads : --l = median plus 1xSD, --u= median plus 3xSD.

## Western blotting

Cell lysates were prepared using RIPA buffer (50 mM Hepes pH 7.9, 140 mM NaCl, 1 mM EDTA, 1% Triton X-100, 0.1% Na-deoxycholate, 0.1% SDS) complemented with sodium pyrophosphate and protease inhibitor cocktail (Sigma). Lysates were cleared by centrifugation and denatured in electrophoresis sample buffer at 95 °C for 5 min. Proteins were separated on 8% Bis-Tris gels and transferred onto a 0.45 µm PVDF membrane (Millipore). Membranes were blocked with 5% skim milk powder in TBS, probed with primary antibodies diluted in 5% BSA in TBS-T and subsequently incubated with the appropriate horseradish peroxidase-coupled secondary antibodies. Visualization was performed using chemiluminescence HRP substrate (Clarity Western ECL Substrate, Bio-Rad).

## Crystal violet staining

For crystal violet staining, MCF10A cells were grown in triplicates on 6-well dishes, fixed with 3.7% paraformaldehyde for 10 min, stained with 0.1% crystal violet (Sigma-Aldrich) in 20% ethanol for 30 min and photographed.

## Co-immunoprecipitation

For exogenous co-immunoprecipitation to analyze homo- and heterodimerization of JUN, 293T cells were transfected using PEI-Max (Sigma-Aldrich). Forty-eight hours after transfection, cells were lysed in RIPA buffer containing protease inhibitor cocktail (Sigma-Aldrich). Immunoprecipitation of Flag-tagged proteins from cleared lysates was performed with Anti-Flag M2 Affinity Agarose Gel (Sigma-Aldrich) at 4 °C for 3 h. Immuno-precipitates were washed three times with RIPA buffer and Flag-tagged proteins were eluted by two consecutive elution steps with 400 µg/ml Flag-peptide (Sigma-Aldrich) for 30 min at 4 °C. Eluates were boiled with sample buffer and subjected to immunoblotting.

## RNA-sequencing

RNA-Sequencing was performed as described previously (Kim et al, 2022). Briefly, total RNA was extracted using RNeasy® Micro Kit (Qiagen) with on-column DNaseI (Qiagen) digestion. RNA integrity (all processed samples had a RIN > 8) was verified with the Agilent Bioanalyzer 2100 automated electrophoresis system (Agilent Technologies). mRNA was isolated using the NEBNext® Poly(A) mRNA Magnetic Isolation Module (NEB) from 1 µg of total RNA and library preparation was conducted with the NEBNext® Ultra RNA Library Prep Kit for Illumina (NEB) with Dual Index Primers (NEBNext® Multiplex Oligos for Illumina, NEB) following the manufacturer's description. Cycles for amplification of the cDNA were determined by qRT-PCR. Libraries were

quantified with the Agilent 2100 Bioanalyzer automated electrophoresis system (Agilent Technologies) and subjected to 75 bp single-end Illumina Sequencing on a NextSeq500. Reads were extracted in FastQ format using bcl2fastq v1.8.4 (Illumina). For all RNA-Sequencing samples, three biological replicates per condition were analyzed.

## RNA-sequencing analysis

Adapter removal, size selection (reads > 25 nt) and quality filtering (Phred score > 43) of FASTQ files were performed with Cutadapt. Reads were then aligned to human genome (hg19) using Bowtie2 (v2.2.9) with default settings. Read count extraction was performed in R using countOverlaps (GenomicRanges).

Differential gene expression analysis was done with DESeq2 (v3.26.8) using default parameters. Cluster analysis of YAP target genes was performed by MFuzz in R using three clusters.

## Statistics and reproducibility

All statistical analyses were performed in R (v4.1.0). The graphs always display the mean value and the standard error of the mean (SEM) unless stated otherwise. The statistical test performed is always given in the respective Figure legend. All experiments were replicated with at least three independent biological replicates unless stated otherwise in the figure legend.

## qRT-PCR

Total RNA was extracted with peqGOLD TriFast Reagent (VWR). First-strand cDNA synthesis was performed from 1 µg RNA using M-MLV Reverse Transcriptase (Promega) and random hexamer primers (Sigma-Aldrich) following the manufacturer's instructions. qPCR reactions were conducted in technical triplicates using InnuMIX qPCR DSGreen Standard Mix (Analytik Jena) supplemented with ROX reference dye on a StepOnePlus™ Real-Time PCR System (Thermo Fisher Scientific). Expression values were normalized to *B2M* as housekeeping gene using the ddCt method. Primer sequences are listed in Reagents and Tools.

## SLAM-Seq

For SLAM-Seq, MCF10A JUN knockout cells expressing JUN-AID-V5 fusion protein and Transport inhibitor response 1 (TIR1) were grown to 60–70% confluency and treated with 100 µM indole-3-acetic acid (IAA, Sigma-Aldrich) for 1 h to deplete JUN protein. Newly synthesized RNA was then labeled by incubating the cells for 1 h with 100 µM 4-thiouridine (4-sU, Sigma-Aldrich). RNA extraction was performed with peqGOLD TriFast Reagent (VWR) and total RNA was alkylated by 10 mM iodoacetamide (Sigma-Aldrich) in PBS at 50 °C for 15 min. Reaction was stopped by adding STOP buffer (100 mM DTT (Merck) in 83.2 mM sodium acetate buffer pH 5.2, 67% ethanol, 20 µg/ml GlycoBlue (Thermo Fisher Scientific)) and incubated 15 min @ −80 °C. RNA was washed with 75% ethanol and resuspended in water. 3′-end mRNA sequencing libraries were generated from 335 ng alkylated RNA using the QuantSeq 3′ mRNA-Seq Library Prep Kit for Illumina (Lexogen). 75 bp single-end sequencing was performed on a NextSeq500 Illumina sequencer.

## SLAM-Seq analysis

SLAM-Seq was analysed by a Nextflow SLAM-Seq pipeline (https://nf-co.re/slamseq). In parallel, a standard RNA-Seq analysis was performed to infer gene expression changes of the steady-state pool. Log2-fold changes of the SLAM-Seq data (based on T-to-C conversions) and the steady-state pool upon auxin-dependent JUN-AID degradation were inferred by DESeq2.

## Gene set enrichment analysis

Gene set enrichment was performed using the MSigDB GSEA tool (v 4.3.1) with a GSEAPreranked analysis.

## Hydrodynamic tail vein injection

Hepatocellular carcinoma was induced in wild-type C57BL/6J mice by delivering Sleeping Beauty (SB) transposon system into the livers of 6-week-old male mice via hydrodynamic tail vein injection (HDTVI). Injection cocktails contained 10 µg pCMV(CAT)T7-SB100 (gift from Zsuzsanna Izsvak; Addgene # 34879); 25 µg pSBbi-Myr-Akt-HA and 25 µg pSBbi-eGFP-IRES-V5-YAP5SA/pSBbi-cJUN-WT-IRES-V5-YAP5SA/pSBbi-cJUN-M14-IRES-V5-YAP5SA. To induce knockdown of NCOR1/2 in the tumors, we first subcloned shRNAs targeting NCOR1/2 in SGEP vector (gift from Christoph Kaether; Addgene #111170). For shNCOR1 and shRen eGFP and mirE cassette were amplified from SGEP, for shNCOR2 only the mirE cassette was amplified from SGEP. Next, polycistronic pSBbi-Myr-Akt-HA-shNCOR1/2 and pSBbi-Myr-Akt-HA-shRen were generated using Gibson assembly of the amplified shRNAs and the bidirectional pSBbi-Myr-Akt-HA construct replacing the original Puromycin resistance. All plasmids were purified using the Endotoxin-free plasmid DNA purification kit (Macherey-Nagel), the total volume was adjusted to 10% (ml) of the body weight (grams) using sterile Ringer´s lactate solution (WDT) and injected into the lateral tail veins of the mice.

## Generation of NIH3T3 NCOR1/2 knockdown cell line

To validate the knockdown efficacy of the polycistronic pSBbi-Myr-Akt-HA-shNCOR1/2 construct, stable NIH3T3 shNCOR1/2 cells were generated by transfection of 1.5 µg of pSBbi-Myr-Akt-HA-shNCOR1/2 and 0.5 µg of pCMV(CAT)T7-SB100. As a control, cells were transfected with pSBbi-Myr-Akt-HA-shRen. After transfection cells were passaged three times, before GFP-positive expressing cells were FACS sorted. After two more passages knockdown efficacy was evaluated by western blot.

## Histology and immunohistochemistry of mouse liver cancer sections

For histological analyses, formalin-fixed paraffin-embedded mouse livers were sectioned at 5 µm and stained with hematoxylin and eosin (H&E). For immunohistochemical analyses, liver sections were stained with the following antibodies using the indicated antigen retrieval method (given in brackets). against the HA-tag (Citrate), YAP (Citrate), JUN (TE), Cleaved-Caspase-3 (Citrate), NCOR1/2 (TE) (Reagents and Tools) following the standard procedures. In brief, after deparaffinization and rehydration,

antigen retrieval was done by boiling the slides in Citrate buffer pH 6.0 (Abcam) or TE buffer pH 9.0 (Abcam). Endogenous peroxygenase was blocked using 3% (v/v) $H_2O_2$ in PBS. After blocking with 5% (w/v) BSA in PBS-T, slides were incubated with primary antibodies in a humidified container at 4 °C overnight. Incubation with appropriate horseradish peroxidase-coupled secondary antibodies was performed at RT for 2 h. Visualization was done using Vector® ImmPACT DAB Peroxidase Substrate (Vector Laboratories), followed by a hematoxylin counterstaining. The slides were imaged using a slide scanner Axio Scan.Z1 microscope (Zeiss).

## Quantification of tumor load

To quantify tumor load from H&E stained sections of mouse livers, the area of all tumor nodules per liver was measured (in pixels) and divided by the total liver area. Quantification was performed using the Trainable Weka Segmentation plugin from FIJI. The analysis was performed in a blinded manner and unblinded after the analysis was complete.

## BioID affinity purification and preparation for MS

MCF10A JUN knockout cells stably expressing BirA*-JUN fusion proteins were treated with 50 μM biotin (Sigma-Aldrich) for 18 h; $1.5 \times 10^7$ cells were collected per sample, snap frozen in liquid nitrogen and stored at −80 °C till further use. Each cell pellet was resuspended in 4.75 ml lysis buffer (50 mM Tris pH 7.5, 150 mM NaCl, 1 mM EDTA, 1 mM EGTA, 1% Triton X-100, 0.1% SDS, 1 mg/ml aprotinin, 0.5 mg/ml leupeptin, 250 U turbonuclease) and rotated for 1 h at 4 °C. The samples were then sonicated (Bioruptor Plus, Diagenode) for 5 cycles (60 s ON/30 s OFF) at high setting and 20 °C. Cell debris was removed by centrifugation (30 min at 4 °C and $17,000 \times g$). Streptavidin Sepharose High Performance Beads (Cytiva) were acetylated by adding 20 mM sulpho-NHS acetate twice for 30 min at RT. The reaction was quenched using 1 M Tris pH 7.5 and the beads were washed extensively with PBS. The acetylated streptavidin beads were equilibrated in lysis buffer, added to the lysate and incubated at 4 °C for 3 h with rotation. After extensive washing with 40 mM ammonium bicarbonate, samples were digested with 1 μg LysC overnight at 37 °C. Peptides were eluted with 150 μl 50 mM ammonium bicarbonate twice and digested with 1 μg trypsin. For elution of the biotinylated peptides, the beads were briefly mixed twice with 150 μl of 80% ACN and 20% TFA. Eluates were dried, resuspended in 200 mM HEPES pH 7.5 and trypsin (1 μg) was added to digest the peptides. Desalting and purification were performed using Waters Oasis® HLB μElution Plate 30 μm (Waters Corporation) according to the manufacturer's instructions. Briefly, the columns were conditioned with $3 \times 100$ μl OASIS Buffer B (80% (v/v) acetonitrile; 0.05% (v/v) formic acid) and equilibrated with $3 \times 100$ μl OASIS Buffer A (0.05% (v/v) formic acid in Milli-Q water). The samples were loaded on the column, washed three times with 100 μl solvent A and then eluted with 50 μl OASIS buffer B twice. Eluates were dried using a speed vacuum centrifuge, resuspended in MS buffer A (5% acetonitrile, 0.1% formic acid) and loaded onto Evotips (Evosep) according to the manufacturer's instructions. Briefly, the Evotips were washed with Evosep buffer B (0.1% formic acid in acetonitrile),

conditioned with 100% isopropanol and equilibrated with Evosep buffer A (0.1% acetonitrile). Subsequently, the samples were loaded onto the Evotips and washed with Evosep buffer A. The loaded Evotips were filled up with buffer A and stored until measurement.

## MS analysis

Peptides were separated using the Evosep One system (Evosep) equipped with an 8 cm × 150 μm i.d. packed with 1.5 μm Reprosil-Pur C18 beads column (Evosep Endurance, EV-1106, PepSep). Samples were run with a pre-programmed proprietary Evosep gradient of 21 min (60 samples per day, 60SPD) with water, 0.1% formic acid, solvent B acetonitrile and 0.1% formic acid as solvents. The LC was coupled to an Orbitrap Exploris 480 (Thermo Fisher Scientific) using PepSep Sprayers and Proxeon nanospray source. The peptides were introduced into the mass spectrometer via a PepSep Emitter 360-μm outer diameter × 20-μm inner diameter, heated to 300 °C, and a spray voltage of 2.2 kV was applied. The temperature of the injection capillary was set to 300 °C and the radio frequency ion funnel to 30%. For DIA data collection, full scan mass spectrometric (MS) spectra with mass range 350–1650 $m/z$ were collected in profile mode in Orbitrap with a resolution of 120,000 FWHM. The default charge state was set to 2+. The fill time was set to a maximum of 45 ms with a limitation of $3 \times 10^6$ ions. DIA scans were recorded with 35 mass window segments of different widths across the MS1 mass range. Higher collisional dissociation fragmentation (stepped normalized collision energy; 25, 27.5 and 30%) was applied and MS/MS spectra were acquired at a resolution of 15,000 FWHM with a fixed first mass of 200 $m/z$ after accumulation of $1 \times 10^6$ ions or after 37 ms filling time (whichever occurred first). Data was collected in profile mode and processed using Xcalibur 4.5 (Thermo Fisher Scientific) and Tune version 4.0.

## Analysis of BioID data

Raw DIA data were analysed using the directDIA pipeline in Spectronaut (v.16, Biognosysis) with BGS settings, except the following parameters: Imputation strategy = Global Imputing, Protein LFQ method = QUANT 2.0, Proteotypicity Filter = Only protein group specific, Major Group Quantity = Median peptide quantity, Minor Group Quantity = Median precursor quantity, Data Filtering = Q-value percentile (0.2), Normalization strategy = Global Normalization on median, Row Selection = Identified in All Runs. Data were searched using a species-specific (*Homo sapiens*, 20.186 entries) and a contaminant database (247 entries) from Swissprot. Data were searched with the following variable modifications: Oxidation (M), Acetyl (Protein N-term), and Biotin (K). A maximum of 2 missed cleavages for trypsin and 5 variable modifications were allowed. Identifications were filtered to achieve a 1% FDR at the peptide and protein levels. Relative quantification was performed in Spectronaut for each paired comparison using the replicate samples from each condition. The data (candidate table) and data reports (protein quantities) were then exported, and further data analyses and visualization was performed with Rstudio using in-house pipelines and scripts. A log2FC cutoff of 0.58 and a q-value < 0.05 were set to select significant proteins.

## CUT&RUN

CUT and RUN experiments were performed as described previously (Kim et al, 2022). Briefly, for each CUT and RUN reaction 200,000 cells were trypsinized, washed, resuspended in 100 μl wash buffer (20 mM HEPES, pH 7.5, 150 mM NaCl, 0.5 mM Spermidine) and bound to 10 μl activated BioMag®Plus Concanavalin A magnetic beads (Polysciences) for 10 min at room temperature. The cells were then incubated with antibodies diluted in 100 μl antibody buffer (wash buffer + 0.01% digitonine and 2 mM EDTA) overnight at 4 °C. IgG rabbit antibody was used as negative control. After incubation with antibodies, the beads were washed in digitonin wash buffer (wash buffer + 0.01% digitonin) and incubated 1 h at 4 °C with 1 μg/ml protein A/G Micrococcal Nuclease fusion protein (pA/G MNase). After washing with digitonin wash buffer, beads were rinsed with low salt buffer (20 mM HEPES, pH 7.5, 0.5 mM Spermidine, 0.01% digitonine), resuspended in 200 μl incubation buffer (20 mM HEPES, pH 7.5, 10 mM CaCl$_2$, 0.01% digitonin) and placed at 0 °C to initiate cleavage. After 30 min, reactions were stopped by adding 200 μl STOP buffer (170 mM NaCl, 20 mM EGTA, 0.01% digitonin, 50 μg/ml RNAse A) and the samples were incubated 30 min at 37 °C to digest the RNA and release the DNA fragments.

The samples were then treated with proteinase K for 1 h at 50 °C and the DNA was purified using Phenol/Chloroform/Isoamyl alcohol. After precipitation with glycogen and Ethanol, the DNA pellet was resuspended in 0.1 X TE and used for DNA library generation with the NEBNext® Ultra™ II DNA Library Prep Kit for Illumina® (New England Biolabs). Adapter ligation was performed with 1:25 diluted adapter and 15 cycles were used for library amplification using dual indices (NEB dual index kit). Paired-end 2 × 25 bp sequencing was performed on a NextSeq500 Illumina Sequencer.

## CUT&RUN analysis

Adapter removal and quality trimming was performed by Cutadapt. Since the carry-over DNA of pAG-MNase purification from *E. coli* was used as spike-in control, mapping was performed to hg19 and a human repeat-masked *E. coli* genome by Bowtie2. Paired-end reads mapped to hg19 with inserts <120 bp were extracted using alignment Sieve (deepTools). A scaling factor was inferred by:

$$\text{scaling factor} = \text{mapped reads}{<}120\,\text{bp to hg19}/\text{mapped reads to } E.\,coli$$

The scaling factor was used to generate spike-in normalized bigWig files by bamCoverage (deepTools). For peak calling, SEACR was used in the "stringent" mode for histone marks, and GoPeaks for transcription factors. The peaks of the individual replicates were intersected by ChIPpeakAnno in R. Only peaks occurring in all replicates were retained for further analysis to generate a conservative peak set. For quantitative analyses, the spike-normalized bigWig files or the output of the bigWigCompare (to compute the log2FCs of two conditions) were used in computeMatrix reference-point (deepTools), e.g., using peaks as reference point. The matrix output of computeMatrix, was then used for further analyses in R. Heatmaps were generated by plotHeatmap (deepTools) using the output of computeMatrix. DiffBind was used to identify differentially bound regions (DBRs). Blacklist and greylist filtered DBRs with a padj < 0.1 were retained for further

analyses. To compute the cumulative distribution frequency (CDF), YAP and JUN signals at YAP peaks were stratified into the following DBR categories (iYAP5SA ON vs. iYAPSA OFF): "YAP only", "JUN only", and "Joint" (YAP and JUN) based on the DiffBind output. The transcriptional start sites (TSSs) were then used as input (-a) in closestBed (BEDtools) to identify the closest YAP peak falling into one of these three categories. Based on this, the cumulation distribution frequency (CDF) for these three peak categories were plotted using ggecdf (ggpubr). The empirical false discovery rate (eFDR) was determined using a Kolgomorov–Smirnov test comparing the "YAP only" peak set (containing the fewest peaks) against 10,000 random peak sets of the same size as the "YAP only" peak set taken from the other two peak categories.

To define enhancers, the peaks for the histone marks H3K27ac and H3K4me1 were intersected with previously published ATAC-Seq experiments performed in iYAP5SA MCF10A cells (Fetiva et al, 2023).

## Data availability

The Next-generation sequencing data generated in this study have been deposited in the GEO database under accession code GSE235968. The microscope images from Fig. 7E,H,L have been deposited in the BioImage Archive under accession code S-BIAD1250.

The source data of this paper are collected in the following database record: biostudies:S-SCDT-10_1038-S44318-024-00188-0.

## Peer review information

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

## Acknowledgements

BvE was supported by grants from the Wilhelm Sander-Stiftung (2022.084.1), BMBF (16GW0271K), DFG (EY 120/4-1), and the German Cancer Aid (Deutsche Krebshilfe; 70113138). YK was supported by Alexander von Humboldt Stiftung. The FLI is a member of the Leibniz Association and is financially supported by the Federal Government of Germany and the State of Thuringia. The DNA Sequencing, the Proteomics, and the Flow Cytometry core facilities as well as the core service histology of the FLI are gratefully acknowledged. We would like to thank all the members of the von Eyss lab, von Maltzahn lab, and Kaether lab for helpful discussion and Tom Hünniger and all the animal care takers at FLI for excellent technical support. We would like to thank Tae-Won Kang for hydrodynamic tail vein injection training. Some of the figures were created with BioRender.com

## Author contributions

**Yuliya Kurlishchuk**: Conceptualization; Formal analysis; Investigation; Visualization; Methodology; Writing—review and editing. **Anita Cindric Vranesic**: Conceptualization; Formal analysis; Investigation; Visualization; Methodology; Writing—original draft; Writing—review and editing. **Marco Jessen**: Conceptualization; Formal analysis; Investigation; Visualization; Methodology; Writing—original draft; Writing—review and editing. **Alexandra Kipping**: Investigation; Visualization. **Christin Ritter**: Investigation. **KyungMok Kim**: Investigation. **Paul Cramer**: Investigation; Methodology. **Björn von Eyss**: Conceptualization; Data curation; Software; Formal analysis; Supervision; Funding acquisition; Investigation; Visualization; Methodology; Writing—original draft; Writing—review and editing.

Source data underlying figure panels in this paper may have individual authorship assigned. Where available, figure panel/source data authorship is listed in the following database record: biostudies:S-SCDT-10_1038-S44318-024-00188-0.

## Funding

## Disclosure and competing interests statement

The authors declare no competing interests.

# Expanded View Figures

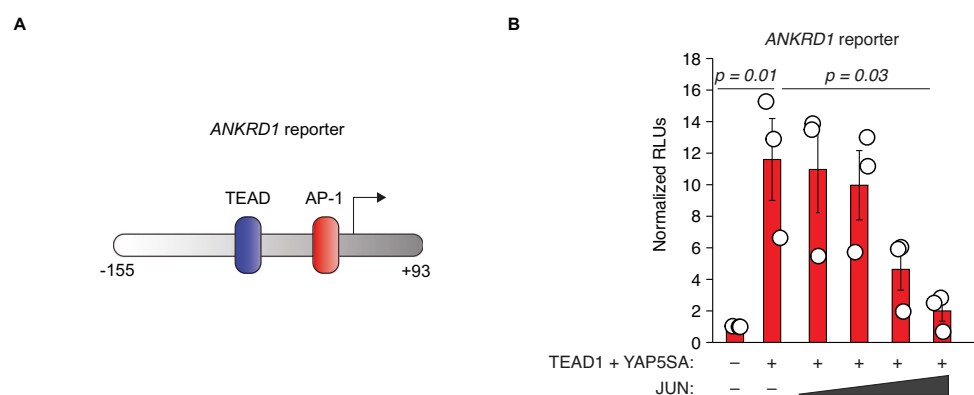

**Figure EV1.  JUN suppresses YAP-dependent induction of the ANKRD1 promoter.**

(**A**) Schematic of the *ANKRD1* reporter construct used for the luciferase reporter assay. (**B**) Luciferase activity of the ANKRD1 reporter in 293T cells co-transfected with vectors for expression of HA-TEAD1 and FLAG-YAP5SA. Increasing amounts of JUN repress the reporter activity. Data shown are from three biological replicates. One-way ANOVA. RLUs = Relative luciferase light units. The error bars indicate the standard error of the mean.

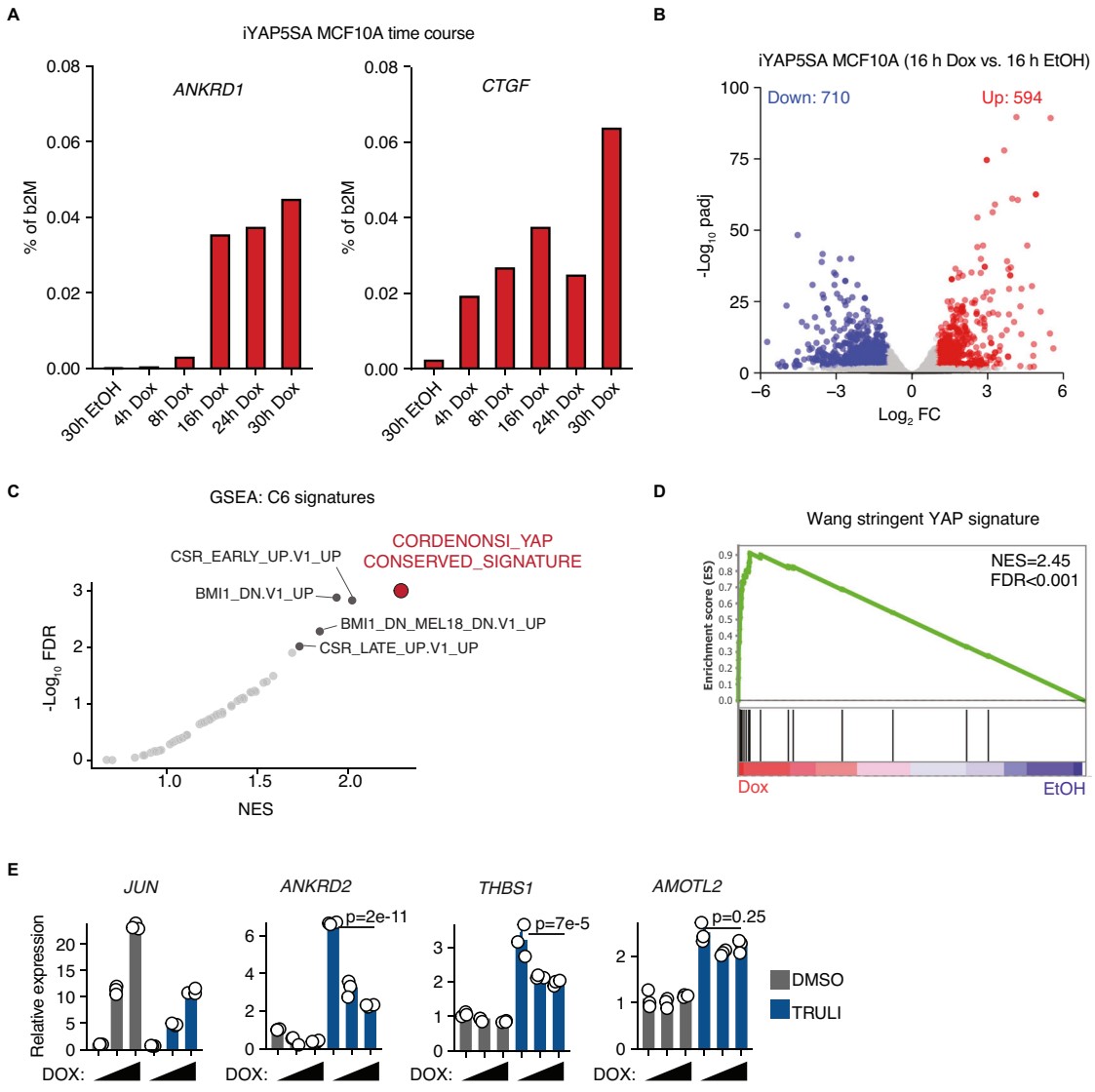

**Figure EV2. The Hippo-insensitive YAP5SA allele induces expression of direct YAP target genes.**

(A) qRT-PCR analysis of YAP target genes, *ANKRD1* and *CTGF*, in iYAP5SA MCF10A cells. YAP5SA expression was induced for indicated times with doxycycline or cells were treated with ethanol as a solvent control. Expression values are shown as percentage of b2M as housekeeping gene. Dox = doxycycline, EtOH = ethanol, n = 1. (B) Volcano plot showing differentially expressed genes upon YAP5SA expression in iYAP5SA MCF10A cells. The cells were treated with either doxycycline or ethanol for 16 h prior to RNA isolation and RNA-Sequencing library preparation. Significantly upregulated genes (determined by DESeq2) are highlighted in red and significantly downregulated genes are highlighted in blue. padj = adjusted *p*-value, FC = fold change. (C) GSEA summary of upregulated gene sets from RNA-Sequencing data upon YAP5SA induction in MCF10A-iYAP5SA cells (n = 3, biological replicates). NES = normalized enrichment score, FDR = false discovery rate. (D) GSEA enrichment plot for the Wang stringent YAP gene signature as a strongly upregulated gene set upon YAP5SA induction in MCF10A-iYAP5SA cells. NES = normalized enrichment score, FDR = false discovery rate. (E) qRT-PCR analysis of YAP target genes in MCF10A cells carrying a doxycycline-inducible JUN allele. JUN expression was induced by treating the cells with increasing concentrations of doxycycline (16.6 ng/ml and 50 ng/ml) or cells were incubated with ethanol as control. Expression of YAP target genes was induced using 10 μM LATS inhibitor TRULI. As control, cells were treated with DMSO. One-way ANOVA. DOX = doxycycline. The error bars indicate the standard error of the mean.

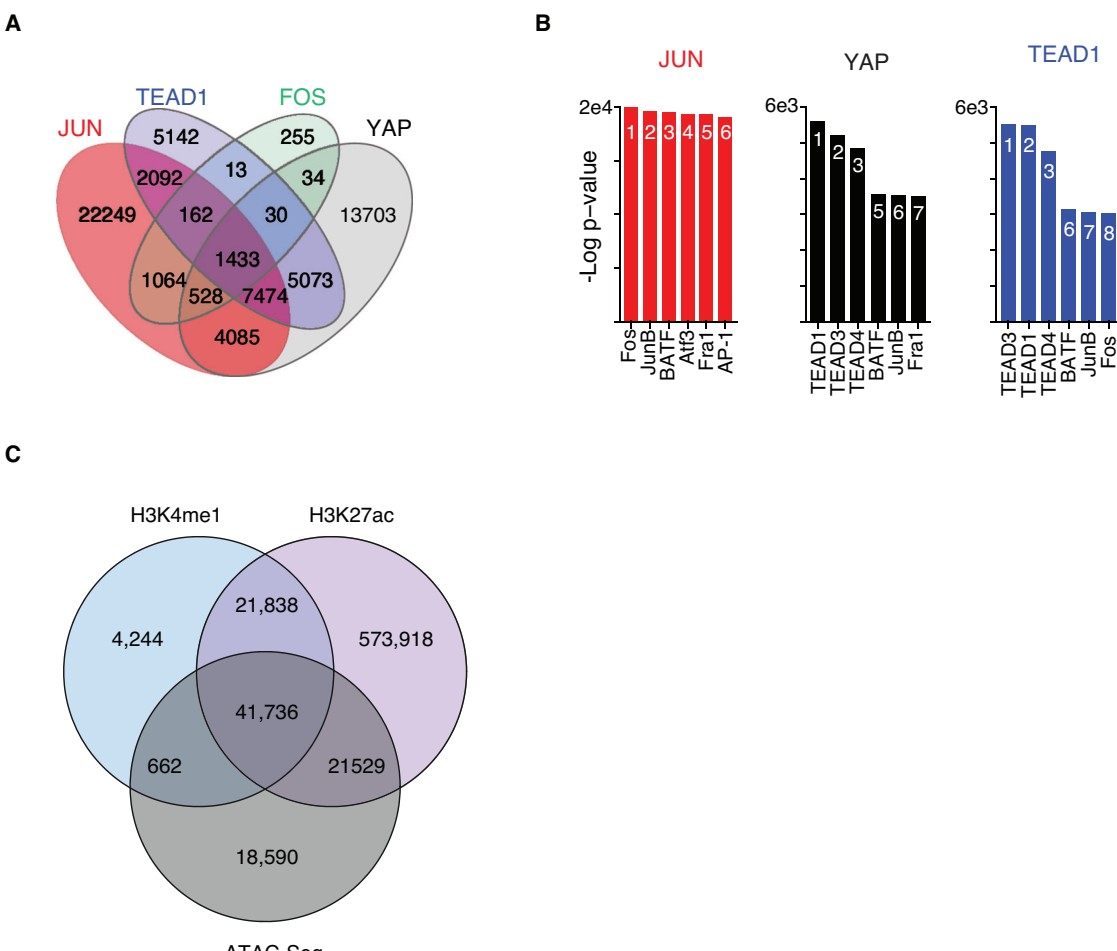

**Figure EV3. JUN is recruited by YAP to genomic sites.**

(**A**) Venn diagram for CUT&RUN peaks in YAP5SA-overexpressing MCF10A cells. (**B**) Motif enrichment analysis (by HOMER) for all CUT&RUN peaks from JUN, YAP, TEAD1. The numbers indicate the rank in the motif analysis. (**C**) Venn diagram for CUT&RUN enhancer marks (H3K4me1 and H3K27ac) and the ATAC-Seq signals.

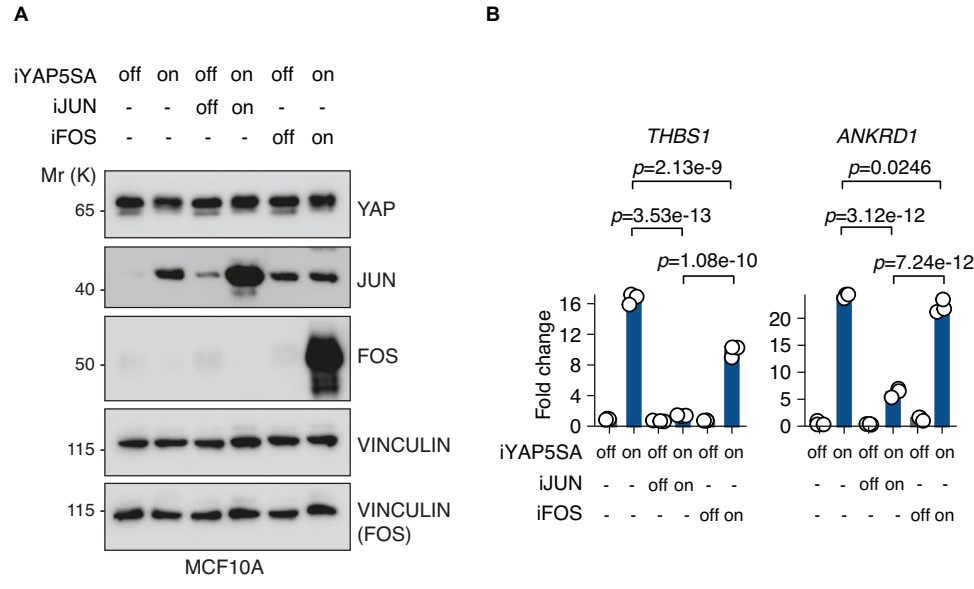

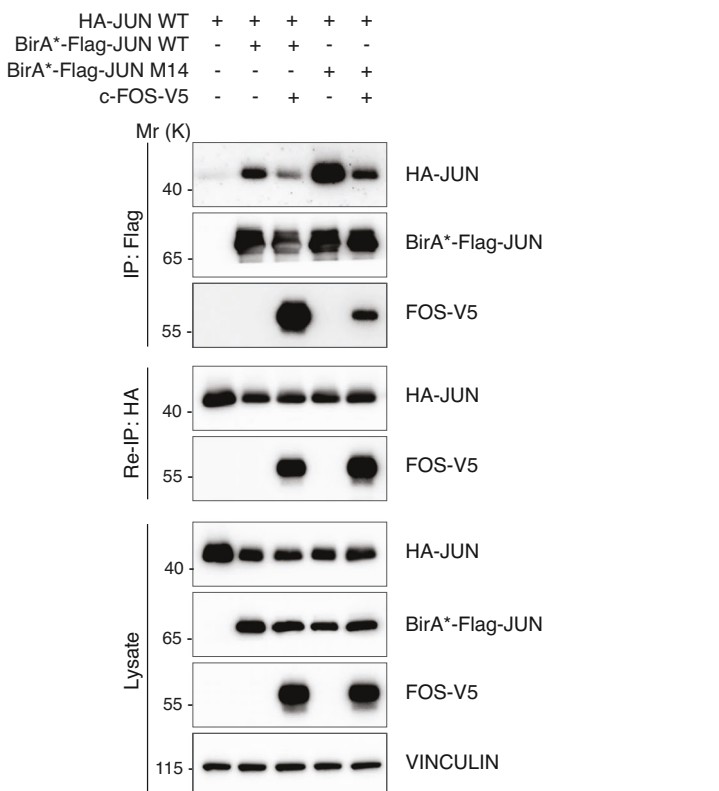

**Figure EV4. JUN-mediated repression of YAP target genes is FOS-independent.**

(A) Immunoblot from inducible MCF10A cells after induction of iYAP5SA, iJUN, and iFOS expression. The cells were treated with 100 ng/ml doxycycline for 20 h prior to analysis. As control, cells were incubated with the same volume of ethanol. VINCULIN was used as loading control, $n = 1$. (B) qRT-PCR analysis of the cells from (A) ($n = 3$, biological replicates). One-way ANOVA. The error bars indicate the standard error of the mean. (C) Co-immunoprecipitation experiments from 293T cells transiently transfected with the indicated constructs. BirA* fusion proteins were immunoprecipitated with Flag, and immunoprecipitates were assayed for HA-JUN and FOS-V5. After Flag precipitation, supernatants were further immunoprecipitated with HA and assayed for FOS-V5, $n = 2$ (biological replicates).

**A**

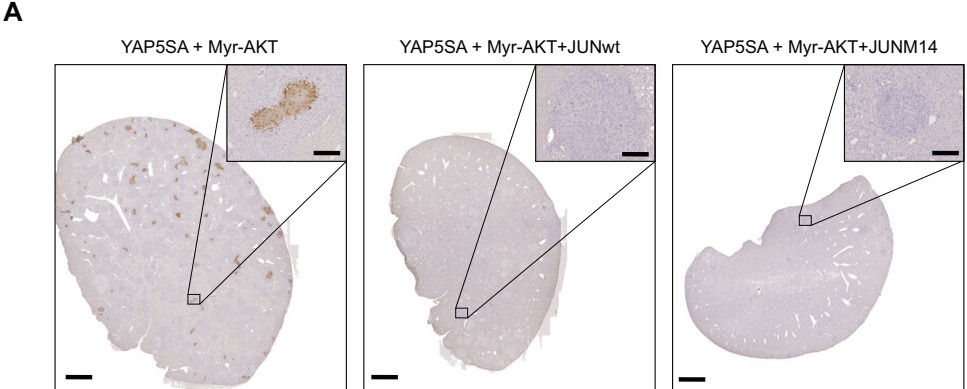

**B**

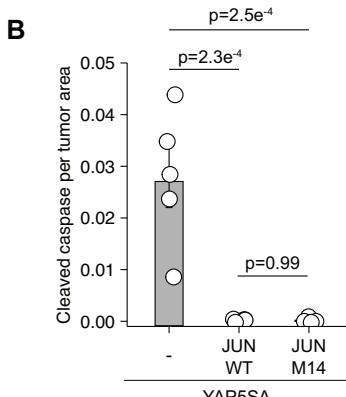

**Figure EV5.  JUN suppresses YAP-induced tumor growth, despite inhibiting apoptosis.**

(A) Representative immunohistochemical staining of cleaved Caspase-3 on HDTVI liver sections. Scale bar = 2 mm. (B) Quantification of cleaved Caspase-3 signals in tumors for the indicated groups. $n = 5$ per experimental group. One-way ANOVA with Tukey HSD post hoc test. One-way ANOVA. The error bars indicate the standard error of the mean.

