## [Peer Review File · The EMBO Journal]

A non-canonical repressor function of JUN restrains YAP activity and liver cancer growth

Yuliya Kurlishchuk, Anita Cindric Vranesic, Marco Jessen, Alexandra Kipping, Christin Ritter, KyungMok Kim, Paul Cramer, and Björn von Eyss

Corresponding author: Björn von Eyss (bjoern.voneyss@leibniz-fli.de)

Review Timeline:

Submission Date:	12th Sep 23
Editorial Decision:	13th Nov 23
Revision Received:	12th Apr 24
Editorial Decision:	20th Jun 24
Revision Received:	3rd Jul 24
Accepted:	12th Jul 24

Editor: Daniel Klimmeck

Transaction Report:

Dear Björn,

Thank you again for submitting your manuscript EMBOJ-2023-115589 for consideration by the EMBO Journal. Please accept my sincere apologies for getting back to you with this unusual protraction due to delayed referee input, as well as detailed discussion in the editorial team. As indicated, your manuscript has been seen by three referees with expertise in Hippo signaling and cancer transcriptional control, and we have received reports from all of them, which are shown below.

As you will see from their comments, the referees acknowledge the potential interest and value of your findings on identification of a non-canonical function of JUN in a JUN-JUN-NCOR1 complex as feedback repressor of YAP-TAZ - driven liver cancer. However, they also express major concerns, which need to be addressed thoroughly to make them supportive of publication in the EMBO Journal. In more detail, referee one is concerned that the mechanistic details of how JUN operates on this non-canonical homodimerized mode as opposed to its homeostatic function- remains too prematurely explored (ref#1, pts. 1,2). This expert also requests corroboration of your model in other cellular contexts and models (ref#1, pts. 3; see also ref#2, pt.2). Reviewer #3 states that that causal involvement of NCOR1/2 in noncanonical JUN cancer function is not conclusively supported in his/her view (ref#3, pt.5). Further, the referees raise a number of points related to additional controls required, data presentation, as well as discussion of the results and terminology used in the manuscript that would need to be conclusively addressed to achieve the level of robustness and clarity needed for The EMBO Journal.

Given the overall interest stated and broader angle of your findings, we are able to invite you to revise your manuscript experimentally to address the referees' comments. I need to stress though that we do require strong support from the referees on a revised version of the study in order to move on to publication of the work.

In light of the extensive experimentation requested, I would appreciate if you could contact me during the next weeks for exchange e.g. a video call to discuss your perspective on the comments and potential plan for revisions.

Please feel free to contact me if you have any questions or need further input on the referee comments.

When submitting your revised manuscript, please carefully review the instructions below.

Please feel free to approach me any time should you have additional questions related to this.

Thank you for the opportunity to consider your work for publication.

I look forward to your revision.

Kind regards,

Daniel Klimmeck

Daniel Klimmeck, PhD
Senior Editor
The EMBO Journal

Instruction for the preparation of your revised manuscript:

- 1) a .docx formatted version of the manuscript text (including legends for main figures, EV figures and tables). Please make sure that the changes are highlighted to be clearly visible.
- 2) individual production quality figure files as .eps, .tif, .jpg (one file per figure).
- 3) a .docx formatted letter INCLUDING the reviewers' reports and your detailed point-by-point response to their comments. As part of the EMBO Press transparent editorial process, the point-by-point response is part of the Review Process File (RPF),

which will be published alongside your paper.

4) a complete author checklist, which you can download from our author guidelines ([https://wol-prod-cdn.literatumonline.com/pb-assets/embo-site/Author Checklist%20-%20EMBO%20J-1561436015657.xlsx](https://wol-prod-cdn.literatumonline.com/pb-assets/embo-site/Author%20Checklist%20-%20EMBO%20J-1561436015657.xlsx)). Please insert information in the checklist that is also reflected in the manuscript. The completed author checklist will also be part of the RPF.

6) It is mandatory to include a 'Data Availability' section after the Materials and Methods. Before submitting your revision, primary datasets produced in this study need to be deposited in an appropriate public database, and the accession numbers and database listed under 'Data Availability'. Please remember to provide a reviewer password if the datasets are not yet public (see <https://www.embopress.org/page/journal/14602075/authorguide#datadeposition>).

7) Our journal encourages inclusion of *data citations in the reference list* to directly cite datasets that were re-used and obtained from public databases. Data citations in the article text are distinct from normal bibliographical citations and should directly link to the database records from which the data can be accessed. In the main text, data citations are formatted as follows: "Data ref: Smith et al, 2001" or "Data ref: NCBI Sequence Read Archive PRJNA342805, 2017". In the Reference list, data citations must be labelled with "[DATASET]". A data reference must provide the database name, accession number/identifiers and a resolvable link to the landing page from which the data can be accessed at the end of the reference. Further instructions are available at .

8) At EMBO Press we ask authors to provide source data for the main and EV figures. Our source data coordinator will contact you to discuss which figure panels we would need source data for and will also provide you with helpful tips on how to upload and organize the files.

Numerical data can be provided as individual .xls or .csv files (including a tab describing the data). For 'blots' or microscopy, uncropped images should be submitted (using a zip archive or a single pdf per main figure if multiple images need to be supplied for one panel). Additional information on source data and instruction on how to label the files are available at .

9) We replaced Supplementary Information with Expanded View (EV) Figures and Tables that are collapsible/expandable online (see examples in <https://www.embopress.org/doi/10.15252/emboj.201695874>). A maximum of 5 EV Figures can be typeset. EV Figures should be cited as 'Figure EV1, Figure EV2' etc. in the text and their respective legends should be included in the main text after the legends of regular figures.

11) For data quantification: please specify the name of the statistical test used to generate error bars and P values, the number (n) of independent experiments (specify technical or biological replicates) underlying each data point and the test used to calculate p-values in each figure legend. The figure legends should contain a basic description of n, P and the test applied. Graphs must include a description of the bars and the error bars (s.d., s.e.m.).

Please remember: Digital image enhancement is acceptable practice, as long as it accurately represents the original data and

conforms to community standards. If a figure has been subjected to significant electronic manipulation, this must be noted in the figure legend or in the 'Materials and Methods' section. The editors reserve the right to request original versions of figures and the original images that were used to assemble the figure.

We realize that it is difficult to revise to a specific deadline. In the interest of protecting the conceptual advance provided by the work, we recommend a revision within 3 months (11th Feb 2024). Please discuss the revision progress ahead of this time with the editor if you require more time to complete the revisions.

Referee #1:

The manuscript by Kurlishchuk, et al reports a c-JUN-mediated negative feedback regulation of the transcriptional complex YAP/TAZ-TEAD in the Hippo pathway. As a component of the AP-1 complex, c-JUN homo-dimerizes to recruit corepressors NCOR1/2 to the TEAD binding sites on genome, thus inhibiting the transcription of YAP downstream genes and YAP-dependent liver growth and cancer development. Since c-JUN expression is also induced by YAP, the authors propose a negative feedback loop for restricting YAP-TEAD transcriptional activity in the nucleus. This study reveals a new function of c-JUN in regulation of the YAP/TAZ transcriptional outputs. However, some of the presented results are quite preliminary requiring more robust data to support the authors' conclusions. Another perceived weakness is that the presented findings are mostly descriptive, lacking mechanistic insights and physiological/pathological relevance.

Main points:

1. Since c-JUN is positively regulated by YAP and required for the YAP-TEAD transcriptional activity, the authors should clarify that to what level of c-JUN upregulation the switch from the c-JUN-dependent positive feedback loop to negative feedback loop will take place? This also raises an issue for the SAM library screen data (Fig1B), because expressing YAP-5SA by viral infection can induce different levels of c-JUN expression in cells, making it hard to interpret the effects.
2. Role of the AP-1 complex in facilitating YAP/TAZ-TEAD downstream gene transcription has been well characterized before. Although the distinct role of c-JUN homodimer reported here is new, key questions should be addressed: 1) what is the mechanism to control the switch between the c-JUN heterodimer and its homodimer? 2) are these two c-JUN dimers exclusive from each other? 3) can c-FOS inhibit c-JUN homodimer formation to sustain the YAP/TAZ-TEAD transcriptional activity? 4) what signaling contexts, what cells/tissues and what organisms is this c-JUN negative feedback loop involved in? 5) is there any pathological connection of this feedback loop with cancers like liver cancer?
3. Regarding the YAP-5SA stable cell line majorly used in the study, first, to increase the significance of the study, the authors should test their hypothesis using other Hippo related models, such as the Hippo KO cells (e.g., LATS1/2 dKO, MOB1 KO), the Hippo inactivating conditions (e.g., serum stimulation, low density); second, the growth-related phenotype of the YAP-5SA MCF10A cells shown here is opposite to that reported in previous studies where YAP-5SA mutant can transform MCF10A cells by promoting cell proliferation in both 2D and 3D cultures, EMT, and anchorage-independent growth. Such discrepancy should be clarified.
4. Key controls are missing to support the proposed model (Fig. 7J). In particular, the c-JUN homodimer mutant and its NCOR-binding mutant should be included for all the studies.

Minor points:

1. In Fig1D, JUND and JUNB were also identified in the SAM screen but not examined here. Are they also involved in the c-JUN-mediated YAP inhibition?
2. Proliferation data shown Fig1A and 1E should be quantified.
3. The manuscript was written in a very concise way. The Introduction and Discussion could be further improved.

Referee #2:

The authors identified JUN as a suppressor of YAP activity in a in vitro genome-wide CRISPR/Cas9 screen, where the positive hits were identified as suppressors of a YAP5SA-driven phenotype. The further analysis of the role of JUN in this phenotype is very meticulous and well-executed. The authors then conclusively show that JUN-JUN homodimers repress YAP-TEAD target genes while JUN-FOS promote expression. These findings are important as they provide completely novel insights into the cooperation of YAP-TEAD and AP1 complexes which drive a large part of the YAP target genes. Certainly, this paper is worth for EMBOJ.

Main comments:

- Most important in my opinion is that the authors make clear that the growth inhibition upon YAP5SA overexpression is a specific YAP effect and not due to some artificial off target effects. Surprisingly, the phenotype of YAP5SA overexpression in the studied cell line (MCF10A) is growth arrest and subsequent overexpression of JUN restores proliferation in YAP5SA oe cells. Typically, YAP hyperactivation results in excessive proliferation, which is what makes it a potential candidate for cancer therapy. The phenotype the authors chose to use for their screen is therefore unusual and the growth arrest upon YAP5SA overexpression could potentially be attributed to artificial effects rather than specific signalling downstream of YAP. However, the authors make good arguments that this phenotype is a specific effect of YAP activity and merely use the growth arrest phenotype as a powerful tool for their screen. Nevertheless, the authors should make this point very explicit in the first sections of the paper. For example, how specific are the genes induced upon YAP5SA overexpression? How does the induced gene set compare to gene sets induced by YAP5SA in other cells? For this a comparison with other published RNA-seq experiments would be informative and sufficient.
- The authors mostly compare MCF10A+YAP5SA+sgNTC and MCF10A+YAP5SA+sgJUN conditions. How do these two conditions compare to normal MCF10A cells? Is proliferation higher in MCF10A+YAP5SA+sgJUN cells than in MCF10A? What about MCF10A+sgJUN cells?
- The authors performed a SLAM-Seq experiment to study the immediate effect on gene transcription after acute JUN depletion. Why would the authors hypothesise that the short and long term effects of JUN KD to be different in terms of its interaction with YAP?
- Using CUT&RUN, the authors show a negative correlation between YAP and JUN chromatin occupancy in joint YAP and JUN peaks. What is the effect of increased JUN and decreased YAP/TEAD occupancy on gene expression? Could the authors combine their chromatin and mRNA readouts to show what is the effect of YAP/JUN overexpression on the transcription mediated by these YAP-dependent JUN enhancers?
- Subsequently, the authors were able to identify an allele of JUN that has the ability to suppress YAP target gene expression, but does not affect canonical AP-1 targets. This allele shows the same chromatin binding specificity as wild type protein, but in terms of protein interactions does not bind FOS, but rather forms JUN::JUN homodimers. This section is clearly written, and the presented data are of high quality. I have no comments here.
- Then, the authors hypothesise that the repressive effects of the YAP/JUN complex can be mediated by one of the repressors interacting with JUN and indeed show that depletion of NCOR1/2 in YAP5SA+JUN cells restores the levels of YAP target THBS1 back to YAP5SA levels. This result is more convincing than the subsequent CUT&TAG of NCOR1 (since it shows higher binding of NCOR1 in YAP/JUN shared peaks in YAP5SA only than in YAP5SA+JUN, which would be the condition with more repression - the opposite to what would be expected). It would be valuable if the authors could show this effect for more than one target, either by Western Blot or RNAseq.
- Finally, the authors use a mouse model of Yap-driven liver cancer to test whether the overexpression of JUN would suppress tumour growth and indeed observe a very clear decrease in tumour load using either WT or M14 alleles of JUN. This experiment shows a striking and unexpected effect that completely corroborates the model of the authors. I have no comments here.

Minor remarks:

- Terms like "sgJUN" are often associated with gene knock-down, rather than overexpression. Maybe the authors could avoid/replace this term to aid the readers.
- in figures the YAP ON and YAP OFF labels are barely distinguishable. Could the authors use something like "ctrl" and "YAP ON" or at least not put the "ON" and "OFF" into superscript?
- line 102 reference should be to Fig2C,D, not Fig1C,D

Referee #3:

In this manuscript, Yuliya et al revealed a mechanism that regulates the interplay between YAP/TAZ and AP-1. They found that

c-JUN, a component of AP-1, specifically represses YAP/TAZ at common target sites, reducing their activity. Interestingly, this repression is independent of canonical AP-1 function. The authors further demonstrated that NCOR1/2 are required for JUN's ability to repress YAP target genes. YAP/TAZ induce c-JUN expression, creating a negative feedback loop that buffers YAP/TAZ activity at common target sites. This feedback loop is disrupted in liver cancer, allowing YAP/TAZ to exert their full oncogenic potential.

However, some concerns should be addressed to make the manuscript more conclusive. Specifically, the following points could improve the manuscript:

1. When using a cell growth model to perform the genetic screening, as well as a tumor mouse study to validate the result in vivo, did the screening results only apply for cell proliferation or were other cellular phenotypes also studied? please clarify
2. It seems that c-JUN interferes with induction of a specific fraction of YAP target genes(Cluster1), which is specifically functioning in HCC. But the expression of cluster2 genes was barely changed. The authors should discuss how JUN selectively repress Yap activities (ie. in cluster1 genes).
3. The author claimed that the c-JUN repression is independent of canonical AP-1 function by using JUNM14, which retains c-JUN DNA binding capacity but cannot activate canonical AP-1 targets. This is an interesting experiment to separate the intertwined JUN functions. Since JUN overexpression achieved a milder recruitment of JUN to the same sites (Fig6c), please also check if JUNM14 also behaved similarly?
4. As the author mentioned, cJUN is involved in HCC development by preventing apoptosis by antagonizing p53 activity. Some apoptosis staining should be performed to investigate this, even though the authors argued that this is dependent on which of the two functions (repress yap or apoptosis) is more important for carcinogenesis.
5. It is interesting to illustrate that NCOR1/2 mediates the repressive role of JUN on yap activity. But the causal relationship between NCOR1/2 on HCC is still weakly established. A NCOR1/2+YAP5SA mouse group would help to further solidify the connection in vivo.
6. The authors suggest that JUN modulation might represent a therapy for HCC. Consider strengthening the claim by inducing JUN expression therapeutically (once HCC has already commenced).

First of all, we would like to thank the reviewers for their constructive criticism and the very helpful suggestions they made in their comments. On this basis, we have now been able to substantially improve our manuscript.

All changes that were made to the text in the manuscript are highlighted in yellow.

Referee #1:

Major:

1. Since c-JUN is positively regulated by YAP and required for the YAP-TEAD transcriptional activity, the authors should clarify that to what level of c-JUN upregulation the switch from the c-JUN-dependent positive feedback loop to negative feedback loop will take place?

We now added a titration using a doxycycline-inducible JUN allele (EV Fig. 2E) and we could demonstrate that repression of YAP target genes (induced by the LATS inhibitor TRULI) is detectable at rather low JUN levels which are ~4x fold above the endogenous. Right now it is very hard to say what determines the putative switch from the activating to repressive JUN functions. In order to account for that, we now removed this argument, and rather refer to a repressive function of JUN as part of a negative feedback loop.

This also raises an issue for the SAM library screen data (Fig1B), because expressing YAP-5SA by viral infection can induce different levels of c-JUN expression in cells, making it hard to interpret the effects.

We agree with the reviewer that interpretation of the effects within the screen are difficult/impossible. Here, we would argue that only those cells were selected where JUN is sufficiently induced by SAM sgRNAs to repress the very high YAP activity induced by lentiviral YAP5SA. Thus, we believe that induction of the endogenous JUN will most likely not contribute much to the screening results since without JUN induction the cells grow very poorly (Fig. 1D,H-J).

2. Role of the AP-1 complex in facilitating YAP/TAZ-TEAD downstream gene transcription has been well characterized before. Although the distinct role of c-JUN homodimer reported here is new, key questions should be addressed:

1) what is the mechanism to control the switch between the c-JUN heterodimer and its homodimer?

Unfortunately, we do not have a perfect answer to this important question yet. Most likely, it is a mixture of different mechanisms contributing to this phenomenon. From our data, it is obvious that such a switch cannot be explained solely by the levels of c-JUN. If this was the case, JUN induction should also lead to repression of "classical" AP-1 targets, such as IL1B, since JUN::JUN homodimers would also be repressive here. However, JUN overexpression led to induction of AP-1 targets which is paralleled by a repression of YAP targets. Thus, we would propose that the repressive function of potential JUN::JUN homodimers is largely instructed by YAP/TEAD at the specific sites, while JUN levels are just one factor contributing to this (see Discussion lines 623-626). However, since the ultimate proof for a JUN::JUN homodimer as the factor defining factor is very difficult, we mostly refer to a FOS-independent function of JUN and added the JUN::JUN homodimers as a potential mediator in the model (Fig. 7N).

2) are these two c-JUN dimers exclusive from each other?

We now performed competitive Co-IPs (EV Fig. 4C) which demonstrate that c-FOS overexpression is able to shift the equilibrium from JUN::JUN homodimers to a mix of JUN::FOS heterodimers and JUN::JUN homodimers.

3) can c-FOS inhibit c-JUN homodimer formation to sustain the YAP/TAZ-TEAD transcriptional activity?

We added experiments using inducible c-FOS (EV Fig. 4A,B). These data demonstrate that c-FOS induction does not repress expression of YAP targets. However, c-FOS induction is certainly not sufficient to counterbalance the repressive impact of JUN. For this reason, we now rather refer to a FOS-independent repressive function of JUN throughout the text, and we propose that (eventually) JUN::JUN homodimers could be the mediators. These data are in line with all our CUT&RUN results as well as the data on the M14 mutant that is defective in FOS binding but retains the ability to recruit NCOR1 (Fig. 6C-E) and to repress YAP target genes (Fig.4D,E).

4) what signaling contexts, what cells/tissues and what organisms is this c-JUN negative feedback loop involved in?

5) is there any pathological connection of this feedback loop with cancers like liver cancer?

We agree with the reviewer to 100% that these two questions are very important and should be addressed in the future. However, to answer which cells/tissues or even organisms the feedback loop is involved in, much broader analyses would be required. The most ideal system to test this, would be the generation of two JUN knockin mouse strains: i) a JUN point mutant that is able to induce JUN targets but is defective in repression of YAP vs. ii) a JUN point mutant that is defective in JUN target gene induction but retains the ability to repress YAP (similar to JUN M14). Since this kind of work would certainly require years of work, we will not be able to address those questions in depth in a revision. However, with the new additional experiments (TRULI: Fig 2L,M [see point 3] and CUT&RUN: Fig. 3E-M), we would argue that this mechanism can be a means to protect an organism from unrestrained YAP/TAZ activity, e.g. occurring in response to low LATS activity. First, induction of YAP target genes through LATS inhibition can be mitigated by JUN (Fig. 2L,M). Second, we now can show that the repressive function of JUN seems to be particularly important at weak enhancers that are normally largely inaccessible (Fig. 3I-M) and not bound by JUN. High YAP activity leads to enhancer invasion of YAP and recruitment of JUN. Thus, we propose that this pathway has evolved to act as a sensor of high YAP activity in order to protect the organism from its detrimental consequences. We also added this aspect in the Discussion (line 612-616).

3. Regarding the YAP-5SA stable cell line majorly used in the study, first, to increase the significance of the study, the authors should test their hypothesis using other Hippo related models, such as the Hippo KO cells (e.g., LATS1/2 dKO, MOB1 KO), the Hippo inactivating conditions (e.g., serum stimulation, low density); second, the growth-related phenotype of the YAP-5SA MCF10A cells shown here is opposite to that reported in previous studies where YAP-5SA mutant can transform MCF10A cells by promoting cell proliferation in both 2D and 3D cultures, EMT, and anchorage-independent growth. Such discrepancy should be clarified. *We now added an experiment using a LATS inhibitor to inactivate both LATS kinases and induce YAP activity. Here, JUN could reduce induction of cluster 1 genes whereas cluster 2 genes remained unaffected (Fig. 2L,M).*

Regarding the growth-related phenotype of the YAP5SA mutant: this was unfortunately not properly explained from our side, and I would like to apologize for this. We now made the point clearer regarding the proliferation defect of cells with constitutive lentiviral overexpression of YAP5SA vs. inducible mild overexpression of YAP5SA.

- 1. We now can demonstrate that, while constitutive overexpression of YAP5SA in MCF10A cells reduced proliferation in monolayer (Fig. 1A,B), the ability to form mammospheres is strongly enhanced (Fig. 1C). Together, it suggests that strong YAP5SA overexpression reduced cell growth in 2D conditions, but does not lead to a general growth defect since the same cells can grow perfectly fine in 3D.*
- 2. To demonstrate the different levels of YAP overexpression (inducible vs. constitutive YAP5SA), we added immunoblots (Fig. 1A).*

3. *To demonstrate the different effects on proliferation in 2D (inducible vs. constitutive YAP5SA), we now added Incucyte growth curves (Fig. 1C,D) showing that: i) the effect of constitutive YAP5SA overexpression on cell proliferation is TEAD-dependent since a YAP5SA mutant with an additional S94A mutation (disrupting the YAP interaction with TEAD) was growing like the empty vector control. ii) the milder YAP5SA overexpression achieved by the inducible system provided an increase in monolayer cell growth, as described by others (Fig. 1B).*

4. Key controls are missing to support the proposed model (Fig. 7J). In particular, the c-JUN homodimer mutant and its NCOR-binding mutant should be included for all the studies.

We performed additional CUT&RUN experiments using the JUN M14 which demonstrate that – just like JUN WT – JUN M14 is able to bind to joint YAP/JUN sites and recruit NCOR1 to these sites (Fig. 6F,I). Since we were not quite sure to what NCOR-binding mutant the reviewer was referring to, we could not perform these experiments.

Minor points:

1. In Fig1D, JUND and JUNB were also identified in the SAM screen but not examined here. Are they also involved in the c-JUN-mediated YAP inhibition?

We now added a panel that demonstrates the ability of all JUN and JUNB family members to restrain YAP activity, whereas JUND is largely defective in this regard (Fig. 2J,K).

2. Proliferation data shown Fig1A and 1E should be quantified.

We added quantifications for the growth defect upon constitutive YAP5SA overexpression (Fig. 1B) and the rescue by SAM sgJUN (Fig. 1I).

3. The manuscript was written in a very concise way. The Introduction and Discussion could be further improved.

We now tried to improve the Discussion taking into account the new results from this revision.

Referee #2:

The authors identified JUN as a suppressor of YAP activity in a in vitro genome-wide CRISPR/Cas9 screen, where the positive hits were identified as suppressors of a YAP5SA-driven phenotype. The further analysis of the role of JUN in this phenotype is very meticulous and well-executed. The authors then conclusively show that JUN-JUN homodimers repress YAP-TEAD target genes while JUN-FOS promote expression. These findings are important as they provide completely novel insights into the cooperation of YAP-TEAD and AP1 complexes which drive a large part of the YAP target genes. Certainly, this paper is worth for EMBOJ.

Main comments:

- Most important in my opinion is that the authors make clear that the growth inhibition upon YAP5SA overexpression is a specific YAP effect and not due to some artificial off target effects. Surprisingly, the phenotype of YAP5SA overexpression in the studied cell line (MCF10A) is growth arrest and subsequent overexpression of JUN restores proliferation in YAP5SA oe cells. Typically, YAP hyperactivation results in excessive proliferation, which is what makes it a potential candidate for cancer therapy. The phenotype the authors chose to use for their screen is therefore unusual and the growth arrest upon YAP5SA overexpression

could potentially be attributed to artificial effects rather than specific signalling downstream of YAP. However, the authors make good arguments that this phenotype is a specific effect of YAP activity and merely use the growth arrest phenotype as a powerful tool for their screen.

Nevertheless, the authors should make this point very explicit in the first sections of the paper. *Regarding the growth-related phenotype of the YAP5SA mutant: this was unfortunately not properly explained from our side, and I would like to apologize for this. We now made the point clearer regarding the proliferation defect of cells with constitutive lentiviral overexpression of YAP5SA vs. inducible mild overexpression of YAP5SA.*

- 1. We now can demonstrate that, while constitutive overexpression of YAP5SA in MCF10A cells reduced proliferation in monolayer (Fig. 1A,B), the ability to form mammospheres is strongly enhanced (Fig. 1C). Together, it suggests that strong YAP5SA overexpression reduced cell growth in 2D conditions, but does not lead to a general growth defect since the same cells can grow perfectly fine in 3D.*
- 2. To demonstrate the different levels of YAP overexpression (inducible vs. constitutive YAP5SA), we added immunoblots (Fig. 1A).*
- 3. To demonstrate the different effects on proliferation in 2D (inducible vs. constitutive YAP5SA), we now added Incucyte growth curves (Fig. 1C,D) showing that: i) the effect of constitutive YAP5SA overexpression on cell proliferation is TEAD-dependent since a YAP5SA mutant with an additional S94A mutation (disrupting the YAP interaction with TEAD) was growing like the empty vector control. ii) the milder YAP5SA overexpression achieved by the inducible system provided an increase in monolayer cell growth, as described by others (Fig. 1B).*

For example, how specific are the genes induced upon YAP5SA overexpression?

How does the induced gene set compare to gene sets induced by YAP5SA in other cells? For this a comparison with other published RNA-seq experiments would be informative and sufficient.

We now performed an additional GSEA for the RNA-Seq data set (YAP5SA ON vs. YAP5SA OFF). Here, the “Cordenonsi YAP conserved signature” was by far the most significant hit. Furthermore, we added a more stringent signature {Wang, 2018, Comprehensive Molecular Characterization of the Hippo Signaling Pathway in Cancer} lacking the proliferation signatures included in the Cordenonsi gene set. The latter gene set was also recently used by the Halder group and shown to be an excellent predictor of YAP/TAZ activity {Kowalczyk, 2022, Hippo signaling instructs ectopic but not normal organ growth}. This gene set was also strongly induced by inducible (i.e. mild) YAP5SA overexpression arguing that the gene sets induced by iYAP5SA are canonical gene sets previously identified by many other labs (EV Fig. 2C,D).

- The authors mostly compare MCF10A+YAP5SA+sgNTC and MCF10A+YAP5SA+sgJUN conditions. How do these two conditions compare to normal MCF10A cells? Is proliferation higher in MCF10A+YAP5SA+sgJUN cells than in MCF10A? What about MCF10A+sgJUN cells?

We now added several Incucyte growth curves (Appendix Fig. S1A-C) which demonstrate that JUN overexpression has no profound effect on cell proliferation in several setting.

- The authors performed a SLAM-Seq experiment to study the immediate effect on gene transcription after acute JUN depletion. Why would the authors hypothesise that the short and long term effects of JUN KD to be different in terms of its interaction with YAP?

We would not really argue that the effects are different. This experiment was rather designed to demonstrate the direct nature of JUN on YAP target genes. Theoretically, it would be possible that JUN overexpression induces a repressor that subsequently inhibits YAP activity. To rule out such an indirect repression, we performed the SLAM-Seq experiment. Since we

can demonstrate that i) induction of de novo transcripts for YAP/TAZ target genes occur only after 2 hours of JUN degradation (Fig. 2O) and ii) in the absence of any changes of the steady state pool (total mRNA), this provides a strong argument against indirect repression. We also added a half sentence in the Results part (lines 241-244) to clarify this.

Using CUT&RUN, the authors show a negative correlation between YAP and JUN chromatin occupancy in joint YAP and JUN peaks. What is the effect of increased JUN and decreased YAP/TEAD occupancy on gene expression? Could the authors combine their chromatin and mRNA readouts to show what is the effect of YAP/JUN overexpression on the transcription mediated by these YAP-dependent JUN enhancers?

We now incorporated three additional replicates for the "YAP5SA ON vs. YAP5SA OFF" condition within the iYAP5SA experimental setup, resulting in a total of five replicates for this specific condition.

Here, we realized that while both previous biological replicates consistently showed such a negative correlation, some of the newer replicates only show a weak or no effect in this regard. Most likely this negative correlation is a secondary effect due to the recruitment of repressive JUN complexes but seem not to be causative for JUN's effect on YAP target genes.

Therefore, we now removed this part of the analysis. However, due to the high number of replicates, we now could perform a differential peak analysis (similar to a differential gene expression (DGE) analysis in RNA-Seq experiments) to identify peaks that show differential binding upon YAP induction (Fig. 3E, F).

This analysis revealed that peaks showing significant recruitment of both YAP and JUN after YAP5SA induction are further away from a transcription start site (TSS) than peaks showing only YAP recruitment (Fig. 3G). Consistently, we can demonstrate that Cluster 1 genes (induced by YAP and repressed by JUN) tend to be further away from the closest YAP peak than Cluster 2 genes (induced by YAP and not affected by JUN). This is consistent with our observation that the "classical" YAP target genes that are analyzed by most labs are unaffected by JUN. This group includes genes like CTGF, CYR61 and AMOTL2. The YAP target gene ANKRD1 was the only exception here since it falls into the Cluster 1 gene category.

Since this analysis pointed towards a specific function of JUN at YAP-regulated enhancers, we performed additional CUT&RUN experiments to map the enhancer landscape in MCF10A cells. In particular, we performed CUT&RUN experiments for enhancer-specific histone modifications, namely H3K4me1 (n=3) and H3K27ac (n=3) and integrated the data with published MCF10A ATAC-Seq data. By overlaying this data with published MCF10A ATAC-Seq data, we were able to identify 41,736 putative enhancer sites in MCF10A cells. After overlaying this data with our CUT&RUN data, we can demonstrate that those enhancers overlapping with JUN peaks that are recruited in a strictly YAP-dependent manner (highlighted in green in Figure 3) have a significantly lower signal for H3K4me1, H3K27ac (Fig. 3J) and chromatin accessibility (Fig. 3K, ATAC) compared to JUN peaks that are detected in all four conditions (highlighted in yellow in Figure 3). This suggests that these YAP-dependent enhancers are rather weak enhancers in an uninduced state. However, these weak enhancer sites show potent recruitment of YAP and JUN (Fig. 3i) as well as a strong increase in chromatin accessibility (Fig. 3K) indicative of an increased enhancer function. Thus, we propose that JUN is able to specifically tame YAP activity at weak enhancers, potentially in order to restrict enhancer invasion induced by oncogenic levels of YAP. Since we believe that this is an important finding of the study, a corresponding section in the discussion was included (lines 612-616).

Subsequently, the authors were able to identify an allele of JUN that has the ability to suppress YAP target gene expression, but does not affect canonical AP-1 targets. This allele shows the same chromatin binding specificity as wild type protein, but in terms of protein interactions

does not bind FOS, but rather forms JUN::JUN homodimers. This section is clearly written, and the presented data are of high quality. I have no comments here.

Thank you very much!!

Then, the authors hypothesise that the repressive effects of the YAP/JUN complex can be mediated by one of the repressors interacting with JUN and indeed show that depletion of NCOR1/2 in YAP5SA+JUN cells restores the levels of YAP target THBS1 back to YAP5SA levels.

This result is more convincing than the subsequent CUT&TAG of NCOR1 (since it shows higher binding of NCOR1 in YAP/JUN shared peaks in YAP5SA only than in YAP5SA+JUN, which would be the condition with more repression - the opposite to what would be expected).

Regarding the point raised by the reviewer "This result is more convincing than the subsequent CUT&TAG of NCOR1 (since it shows higher binding of NCOR1 in YAP/JUN shared peaks in YAP5SA only than in YAP5SA+JUN, which would be the condition with more repression". I assume, here the reviewer misread the labelling since the third condition was NCOR1 recruitment induced by JUN overexpression in the absence of YAP induction (YAP OFF). Most likely, this was due to the tiny labelling (also mentioned by Reviewer #2) which we now modified. Furthermore, we now used a unified way to plot these analyses to make it easier digestible for the reader. Now, we always plotted the signal ratios (YAP5SA ON/OFF, JUN/FOS, JUN oe/ev) as heatmaps (Fig. 6C) and as histograms (Fig. 6D).

It would be valuable if the authors could show this effect for more than one target, either by Western Blot or RNAseq.

Meanwhile, we generated NCOR1 CRISPR knockout clones which show the same effect as the siRNAs, and included additional target genes. We tried to generate NCOR1/2 DKO cells, but, unfortunately, we were not successful in obtaining DKO cells despite screening dozens of clones. However, the analysis shows that deletion of NCOR1 is sufficient to trigger a superinduction of YAP cluster 1 target genes whereas cluster 2 genes are unaffected (Fig. 6B,C).

- Finally, the authors use a mouse model of Yap-driven liver cancer to test whether the overexpression of JUN would suppress tumour growth and indeed observe a very clear decrease in tumour load using either WT or M14 alleles of JUN. This experiment shows a striking and unexpected effect that completely corroborates the model of the authors. I have no comments here.

Thank you very much!!

Minor remarks:

- Terms like "sgJUN" are often associated with gene knock-down, rather than overexpression. Maybe the authors could avoid/replace this term to aid the readers.

We now modified the labelling to SAM sgJUN etc.

- in figures the YAP ON and YAP OFF labels are barely distinguishable. Could the authors use something like "ctrl" and "YAP ON" or at least not put the "ON" and "OFF" into superscript?

We modified the labelling to iYAP5SA OFF and iYAP5SA ON omitting superscript.

- line 102 reference should be to Fig2C,D, not Fig1C,D

We corrected this mistake, thank you.

Referee #3:

In this manuscript, Yuliya et al revealed a mechanism that regulates the interplay between YAP/TAZ and AP-1. They found that c-JUN, a component of AP-1, specifically represses YAP/TAZ at common target sites, reducing their activity. Interestingly, this repression is independent of canonical AP-1 function. The authors further demonstrated that NCOR1/2 are required for JUN's ability to repress YAP target genes. YAP/TAZ induce c-JUN expression, creating a negative feedback loop that buffers YAP/TAZ activity at common target sites. This feedback loop is disrupted in liver cancer, allowing YAP/TAZ to exert their full oncogenic potential.

However, some concerns should be addressed to make the manuscript more conclusive. Specifically, the following points could improve the manuscript:

1. When using a cell growth model to perform the genetic screening, as well as a tumor mouse study to validate the result in vivo, did the screening results only apply for cell proliferation or were other cellular phenotypes also studied? please clarify

Regarding the growth-related phenotype of the YAP5SA mutant: this was unfortunately not properly explained from our side, and I would like to apologize for this. We now made the point clearer regarding the proliferation defect of cells with constitutive lentiviral overexpression of YAP5SA vs. inducible mild overexpression of YAP5SA.

- 1. We now can demonstrate that, while constitutive overexpression of YAP5SA in MCF10A cells reduced proliferation in monolayer (Fig. 1A,B), the ability to form mammospheres is strongly enhanced (Fig. 1C). Together, it suggests that strong YAP5SA overexpression reduced cell growth in 2D conditions, but does not lead to a general growth defect since the same cells can grow perfectly fine in 3D.*
- 2. To demonstrate the different levels of YAP overexpression (inducible vs. constitutive YAP5SA), we added immunoblots (Fig. 1A).*
- 3. To demonstrate the different effects on proliferation in 2D (inducible vs. constitutive YAP5SA), we now added Incucyte growth curves (Fig. 1C,D) showing that: i) the effect of constitutive YAP5SA overexpression on cell proliferation is TEAD-dependent since a YAP5SA mutant with an additional S94A mutation (disrupting the YAP interaction with TEAD) was growing like the empty vector control. ii) the milder YAP5SA overexpression achieved by the inducible system provided an increase in monolayer cell growth, as described by others (Fig. 1B).*

2. It seems that c-JUN interferes with induction of a specific fraction of YAP target genes (Cluster1), which is specifically functioning in HCC. But the expression of cluster2 genes was barely changed. The authors should discuss how JUN selectively repress Yap activities (ie. in cluster1 genes).

We now incorporated three additional replicates for the "YAP5SA ON vs. YAP5SA OFF" condition within the iYAP5SA experimental setup, resulting in a total of five replicates for this specific condition.

Here, we realized that while both "old" replicates consistently showed such a negative correlation, some of the newer replicates only show a weak or no effect in this regard. Most likely this negative correlation is a secondary effect due to the recruitment of repressive JUN complexes but most likely are not causative for JUN's effect on YAP target genes.

Therefore, we now removed this part of the analysis. However, due to the high number of replicates, we now could perform a differential peak analysis (similar to a differential gene

expression (DGE) analysis in RNA-Seq experiments) to identify peaks that show differential binding upon YAP induction (Fig. 3E, F).

This analysis revealed that peaks showing significant recruitment of both YAP and JUN after YAP5SA induction are further away from a transcription start site (TSS) than peaks showing only YAP recruitment (Fig. 3G). Consistently, we can demonstrate that Cluster 1 genes (induced by YAP and repressed by JUN) tend to be further away from the closest YAP peak than Cluster 2 genes (induced by YAP and not affected by JUN). This is consistent with our observation that the “classical” YAP target genes that are analyzed by most labs are unaffected by JUN. This group includes genes like CTGF, CYR61 and AMOTL2. The YAP target gene ANKRD1 was the only exception here since it falls into the Cluster 1 gene category.

Since this analysis pointed towards a specific function of JUN at YAP-regulated enhancers, we performed additional CUT&RUN experiments to map the enhancer landscape in MCF10A cells. In particular, we performed CUT&RUN experiments for enhancer-specific histone modifications, namely H3K4me1 (n=3) and H3K27ac (n=3) and integrated the data with published MCF10A ATAC-Seq data. By integrating this data with published MCF10A ATAC-Seq data, we were able to identify 41,736 putative enhancer sites in MCF10A cells. After overlaying this data with our CUT&RUN data, we can demonstrate that those enhancers overlapping with JUN peaks that are recruited in a strictly YAP-dependent manner (highlighted in green in Figure 3) have a significantly lower signal for H3K4me1, H3K27ac (Fig. 3J) and chromatin accessibility (Fig. 3K, ATAC) compared to JUN peaks that are detected in all four conditions (highlighted in yellow in Figure 3). This suggests that these YAP-dependent enhancers are rather weak enhancers in an uninduced state. However, these weak enhancer sites show potent recruitment of YAP and JUN (Fig. 3i) as well as a strong increase in chromatin accessibility (Fig. 3K) indicative of an increased enhancer function. Thus, we propose that JUN is able to specifically tame YAP activity at weak enhancers, potentially in order to restrict enhancer invasion induced by oncogenic levels of YAP. Since we believe that this is an important finding of the study, a corresponding section in the discussion was included (lines 564-570).

3. The author claimed that the c-JUN repression is independent of canonical AP-1 function by using JUNM14, which retains c-JUN DNA binding capacity but cannot activate canonical AP-1 targets. This is an interesting experiment to separate the intertwined JUN functions. Since JUN overexpression achieved a milder recruitment of JUN to the same sites (Fig6c), please also check if JUNM14 also behaved similarly?

We performed additional CUT&RUN experiments as requested by the reviewer (Fig. 6). Here, it becomes evident that JUN WT as well as JUN M14 overexpression lead to a very mild recruitment at joint YAP/JUN sites (Fig. 6F,I) when compared to the JUN recruitment achieved by YAP induction (Fig. 6D,G). This argues that YAP is the limiting/determining factor with regard to JUN recruitment at these sites.

4. As the author mentioned, cJUN is involved in HCC development by preventing apoptosis by antagonizing p53 activity. Some apoptosis staining should be performed to investigate this, even though the authors argued that this is dependent on which of the two functions (repress yap or apoptosis) is more important for carcinogenesis.

As suggested by the reviewer, we performed cleaved Caspase-3 staining on sections from the three different conditions (EV Fig. 5A,B). Here, we were able to detect apoptosis in the “YAP5SA only” tumors but we failed to detect any signs of apoptosis in the YAP5SA+JUN WT or the YAP5SA+JUN M14 conditions. Since the conditions with JUN overexpression show potently reduced tumor growth, this reduced tumor growth cannot be explained by JUN’s effect on apoptosis. The absence of apoptosis would rather work against the JUN-dependent phenotypes described here.

5. It is interesting to illustrate that NCOR1/2 mediates the repressive role of JUN on yap activity. But the causal relationship between NCOR1/2 on HCC is still weakly established. A NCOR1/2+YAP5SA mouse group would help to further solidify the connection in vivo.

We performed additional mouse experiments on a cohort of 28 mice using the hydrodynamic tail vein injections (HDTV) HCC model in conjunction with a mirE-based tandem shRNA targeting NCOR1/2. We conducted HDTV experiments for three experimental groups: 1.) YAP5SA+shRen, 2.) YAP5SA+JUN+shRen and 3.) YAP5SA+JUN+shNCOR1/2. Unfortunately, the tandem NCOR1/2 shRNAs led to a rather weak knockdown efficiency but there was a clear trend of restored tumor growth in the shNCOR1/2 condition (Fig.7I-K). Nevertheless, we noticed something very striking when we analyzed the tumors for NCOR1 and NCOR2 expression: most tumors that arose in the YAP5SA + JUN + non-targeting shRenilla control had already lost NCOR1 and NCOR2 expression (Fig. 7K,L) whereas NCOR1/2 expression was readily detectable in the YAP5SA + shRen control (without JUN overexpression).

This suggests the conclusion that tumors overexpressing JUN can only develop when they lose NCOR1/2 expression, either through downregulation or mutation, thus diminishing the repressive effect of JUN on YAP target genes, as illustrated in Figure 6. The modest knockdown of NCOR1/2 may accelerate tumor development in those tumors expressing sufficient levels of NCOR1/2 shRNAs, which have not yet lost NCOR1/2 expression through alternative mechanisms.

In summary, this unexpected but unbiased finding reinforces the association between NCOR1/2 and JUN's capacity to suppress YAP target genes and demonstrates a role in YAP-dependent liver cancer.

6. The authors suggest that JUN modulation might represent a therapy for HCC. Consider strengthening the claim by inducing JUN expression therapeutically (once HCC has already commenced).

We agree that it would be ideal to strengthen this claim using an inducible model to inactivate NCOR1/2 in established HCCs. However, given the weak knockdown efficiencies we achieved with constitutive shRNA, we have refrained from performing this experiment. However, to address this point, we now removed this part of the discussion.

Dear Björn,

Thank you for submitting your revised manuscript (EMBOJ-2023-115589R) to The EMBO Journal. Please accept my apologies for the unusual delay in the process. Your amended study was sent back to the three referees for their scientific re-evaluation, and we have received detailed comments from two of them, which I enclose below. Please note that while reviewer #3 was not able at this time to reassess your work, we have asked referee #2 for his-her input on whether the critique was addressed satisfactorily. As you will see from the comments enclosed below, the expert states that the work has been substantially improved by the revisions and s/he is now in favour of publication. We note that referee #1 remains more critical, however given the strong support by referees #2 and #3 over the two rounds of assessment, we now concluded that we can move forward with this work towards publication.

Thus, we are pleased to inform you that your manuscript has been accepted in principle for publication in The EMBO Journal.

We now need you to take care of a number of issues related to formatting and data presentation as detailed below, which should be addressed at re-submission.

Please contact me at any time if you have additional questions related to below points.

As you might have seen on our web page, every paper at the EMBO Journal now includes a 'Synopsis', displayed on the html and freely accessible to all readers. The synopsis includes a 'model' figure as well as 2-5 one-short-sentence bullet points that summarize the article. I would appreciate if you could provide this figure and the bullet points.

Thank you for giving us the chance to consider your manuscript for The EMBO Journal. I look forward to your final revision.

Again, please contact me at any time if you need any help or have further questions.

Best regards,

Daniel

>> Author Contributions: Please remove the author contributions information from the manuscript text. Note that CRediT has replaced the traditional author contributions section as of now because it offers a systematic machine-readable author contributions format that allows for more effective research assessment. and use the free text boxes beneath each contributing author's name to add specific details on the author's contribution.

More information is available in our guide to authors.
<https://www.embopress.org/page/journal/14602075/authorguide>

>> Adjust the title of the 'Competing Interests' section to 'Disclosure and Competing Interests Statement' and move after Acknowledgements.

>> Appendix: the appendix needs a table of contents including page numbers added, correct nomenclature is "Appendix Figure S1" etc. .

>> Funding: please mention the 'Alexander von Humboldt Stiftung and FLI support by Federal Government of Germany and the State of Thuringia' in our online system.

>> Please provide source data for the study as to the separate request e-mail by my colleague Hannah Sonntag.

>> Data availability section: remove referee token and make sure privacy is released form the online dataset.

>> Dataset EV legends: Table EV 1-5 should be renamed Dataset EV1 - 5 and each file needs a legend added in a separate tab. For Table EV6 please use the template format and rename it "Reagents and Tools".

>> Consider additional changes and comments from our production team as indicated below:

- Figure legends:

1. Please define the annotated p values *** in the legend of figure 3g; as appropriate.
2. Please indicate the statistical test used for data analysis in the legends of figures 1f, i; 3g; EV 2b; EV 3b.
3. Please note that the box plots need to be defined in terms of minima, maxima, centre, bounds of box and whiskers, and percentile in the legends of figures 2p; 3h, l-m; 4h.
- 4.. Please note that information related to n is missing in the legends of figures 2h, m, p; 3d, h, l-m; 4e, h; 6c; 7f; EV 2c; EV 4b.
5. Please note that n=2 in figure 2k.
6. Although 'n' is provided, please describe the nature of entity for 'n' in the legends of figures 7g, k.
7. Please note that the error bars are not defined in the legends of figures 1b-c; 2d, h, k, m; 4b, e; 6c; 7f-g, k, m; EV 1b; EV 2c; EV 4b; EV 5b.
8. Please note that the scale bar needs to be defined for figure 7h.

Referee #1:

In the revised manuscript, the authors performed additional experiments and addressed some of the questions I raised before. However, the mechanism of the activating- and inactivating- JUN complex switch and the physiological relevance of the JUN-dependent negative feedback loop in Hippo signaling are still missing. These weaknesses make the presented findings relatively descriptive, somehow reducing the significance of this study.

Referee #2:

The authors satisfactorily addressed my questions.

Referee #2, additional comments on authors' response to referee #3:

In my opinion, the authors addressed the concerns of reviewer 3. I am not sure they really answered what s/he asked in question 1 but I do not think that this is a problem. I suggest to accept the paper.

The authors addressed the minor editorial issues.

Dear Dr. von Eyss,

Thank you for submitting the revised version of your manuscript. I have now evaluated your amended manuscript and concluded that the remaining minor concerns have been sufficiently addressed.

I am thus pleased to inform you that your manuscript has been accepted for publication in the EMBO Journal.

On a different note, I would like to alert you that EMBO Press offers a format for a video-synopsis of work published with us, which essentially is a short, author-generated film explaining the core findings in hand drawings, and, as we believe, can be very useful to increase visibility of the work. Please see the following link for representative examples and their integration into the article web page:

<https://www.embopress.org/doi/full/10.15252/emj.2019103932>

Finally, we have noted that the submitted version of your article is also posted on the preprint platform bioRxiv. We would appreciate if you could alert bioRxiv on the acceptance of this manuscript at The EMBO Journal in order to allow for an update of the entry status. Thank you in advance!

Best regards,

Daniel Klimmeck

Daniel Klimmeck, PhD
Senior Editor
The EMBO Journal
EMBO
Postfach 1022-40
Meyerohofstrasse 1
D-69117 Heidelberg
contact@embojournal.org
Submit at: <http://emboj.msubmit.net>
